

# Technical note: TEOS-10 EXCEL - Implementation of the Thermodynamic Equation Of Seawater - 2010 in EXCEL

Carlos Gil Martins[1,2], Jaimie Cross[1,2]

[1]MLA College, The Merchant, St Andrew Street, Plymouth PL1 2AX, UK
[2]Faculty of Engineering and Natural Sciences, Bahçeşehir University, Istanbul, Turkey

*Correspondence to*: Carlos G. Martins (carlos.martins@mla-uk.com)

**Abstract.** This paper and associated software implement the Thermodynamic Equation Of Seawater - 2010 (TEOS-10) in EXCEL for an expedite estimation of *absolute salinity* ($S_A$), *conservative temperature* ($\Theta$) and derived thermodynamic properties of seawater – *potential density* ($\sigma_\Theta$), *in situ density* ($\rho_{S_A, \Theta, p}$) and *sound speed* ($c$). Vertical profile template plots for

these parameters are included, as well as a $S_A$ - $\Theta$ diagram template, which includes plotting of the density field (computation of user selected $\sigma_\Theta$ lines is included). Estimation of *absolute salinity* relies on the interpolation of data from casts of seawater from the world ocean (IOC, SCOR and IAPSO, 2010), and the EXCEL workbook introduced here (TEOS-10 EXCEL, available at https://doi.org/10.5281/zenodo.4763574) includes a subset of the TEOS-10 look-up tables necessary for this estimation, namely the *salinity anomaly* [deltaSA_ref] and the *absolute salinity anomaly ratio* [SAAR_ref] look-up tables. As

the user simply needs to paste new data into the spreadsheet to automatically compute the oceanographic parameters referred above, this tool may prove to be extremely useful among all who are not comfortable of using the full-featured TEOS-10 programming language environments (e.g., MATLAB, FORTRAN), but rather need a simpler way of computing fundamental properties of seawater (e.g., density, sound speed), while adhering to current standards. Returned values are the same (up to 15 decimal places, i.e., difference = 0.000000000000000), as the ones obtained with the MATLAB version of the GSW (Gibbs

Sea Water) toolbox (McDougall and Barker, 2011) available at the TEOS-10 website (https://www.teos-10.org). This paper describes the EXCEL workbook, its use, and the included VBA (Visual Basic for Applications) functions. Quality control against the GSW toolbox is also addressed, namely issues detected with the interpolated values returned by the toolbox when there are missing values in the reference look-up table. In these situations, the GSW toolbox replaces missing values with a level pressure horizontal interpolation of neighbour points, while it is clear from the testing results that vertical interpolation,

which was then implemented in TEOS-10 EXCEL, returns a more robust solution (figs. 9 and 10).

## 1. Introduction

The development of software to facilitate the efficient analysis of the properties of seawater has allowed users to better understand the marine environment, assisting members of the student, research, and industrial communities alike. One such initiative, the Gibbs Sea Water (GSW) Toolbox (McDougall and Barker, 2011), implements the Thermodynamic Equation of





Seawater – 2010 (TEOS-10) into software that rapidly calculates required seawater properties through utilisation of the MATLAB programming language. However, the software requires a working understanding and knowledge of MATLAB and of programming generally. As such, the toolbox may not be as readily accessible to all practitioners within the field of marine data analysis (e.g., Buzzetto-More et al., 2010; Bosse and Gerosa, 2017). The aim of this paper is to present an implementation of TEOS-10, within Microsoft EXCEL, a popular and readily available application. This new implementation requires no

specialist knowledge to operate; it is therefore hoped that all groups interested in analysing sea water properties may benefit from free and open access to this new tool.

Seawater can be defined as a thermodynamic system with one liquid phase and two components: i) pure water and ii) dissolved salts. At the end of the XIX century, J. Willard Gibbs, established the *Gibbs phase rule* (Gibbs, 1874–1878) which states that,

for a multiphase system in thermodynamic equilibrium, such as seawater, the degrees of freedom of the system, i.e., the number of independent variables needed to define it, equals the number of components subtracted by the number of phases plus two. For seawater this adds to three (2-1+2) and the 'chosen' variables are Salinity, Temperature and Pressure. The properties related to the three variables must be conservative, i.e., must not globally change within the system (i.e., the ocean), by opposition to non-conservative properties that are created and consumed within it (e.g., oxygen).


If the concept of pressure and temperature have remained pretty much unaltered over time (although the temperature standard changed in 1989 from IPTS-68 to ITS-90 (Preston-Thomas, 1990)), the definition of salinity has suffered significative variations during the last century (Millero, 2010). The current Thermodynamic Equation Of Seawater - 2010 (TEOS-10) has introduced a new salinity quantity, *absolute salinity* ($S_A$), that can be defined as "the mass fraction of dissolved material in

seawater" (IOC, SCOR and IAPSO, 2010: 3). Accompanying $S_A$, a new temperature quantity, *conservative temperature* ($\Theta$), was also introduced. *Conservative temperature* is estimated from *potential temperature* ($\theta$) and $S_A$ and is two orders of magnitude more conservative than $\theta$ (IOC, SCOR and IAPSO, 2010: 5). These two new quantities, $S_A$ and $\Theta$, together with *pressure* ($p$), are now the arguments of the equation of state, and to compute any thermodynamic property of seawater (e.g., density, sound speed) they must be estimated first. *Practical salinity* ($S_P$), which was used before in the Equation of State – 80

(EOS-80), is however still required for the determination of $S_A$ and remains being the salinity quantity recommended to be archived in oceanographic data bases (IOC, SCOR and IAPSO, 2010: 8).

The polynomial nature of EOS-80 allowed the easy implementation of algorithms for computation of seawater properties, which led to the proliferation of stand-alone applications, interactive web sites and Visual Basic for Applications (VBA)

modules. In TEOS-10 however, *absolute salinity* can be only estimated from interpolation of measured absolute salinity anomalies stored in a world atlas look-up table. This difficulty might be a possible explanation for the absence of any previous implementation of TEOS-10 in EXCEL. Section 2 introduces the new workbook, explains its operation, and describes the access to the world ocean look-up tables. Section 3 describes the translation of the original MATLAB code into VBA and





discusses the interpolation method used for missing data in the reference look-up table, followed by the conclusion and
summary in Section 4.

## 2. The TEOS-10 EXCEL Workbook

An EXCEL workbook file that implements a sub-set of the GSW (Gibbs Sea Water) toolbox (available at the TEOS-10 website
https://www.teos-10.org) accompanies this paper. The file includes sample data that can be easily replaced by new user data
to obtain ocean vertical profiles and $S_A - \Theta$ diagrams. The computation algorithms are implemented as VBA functions and are
used as any other standard EXCEL function. The TEOS-10 world ocean look-up tables of measured *absolute salinity anomaly*
[deltaSA_ref] and *absolute salinity anomaly ratio* [SAAR_ref], essential to estimate *absolute salinity*, are included in the
workbook and are described later in the paper. The desktop App version of Microsoft Office is needed to use the workbook,
as *VBA Macros do not run in Microsoft Web Office*. On opening the EXCEL file, authorisation for running macros must be
granted.


The workbook (Fig. 1) contains four data spreadsheets (three green tabs and one yellow), two plotting spreadsheets (blue tabs)
and six TEOS-10 look-up tables (purple tabs). Pressing [Alt – F11] opens the VBA environment allowing access to the 13
function modules, although access to these is not required to make use of the Workbook, nor is a working knowledge of VBA.

### 2.1 The green data tabs

The structure of the three green data tabs is identical, the only difference being the data sets incorporated in each. The 'TEOS-
10 Test Data' spreadsheet includes a testing data set from the GSW Toolbox, located in the NW Pacific at 162.5º E 33º N.
'TS-55' data is a 1º longitude x 1º latitude historical average vertical profile in the NE Atlantic, off the Iberian Peninsula,
centred at 10.5º W 40.5º N, with pressure levels interpolated to standard 'Levitus' levels (Levitus, 1982), and 'CTD-020' is a
CTD cast in the same grid bin, at 10º 01º W 40º 05' N (Martins, 1998).  Seawater properties in coloured columns are computed
85    on the fly from user data input in white cells. The data included (in white cells) can be replaced by user data. Spreadsheet lines
can be added (or deleted) and, if additional lines are required, it would only be needed to copy down the coloured cells for the
formulae to propagate over the extra lines, without any further adjustments being necessary. The only caution users should
have, is to *not move* the data (white cells) to other locations, as the spreadsheet formulae will 'follow' this operation, disrupting
the original cell referencing. Users may also add new data spreadsheets to the EXCEL workbook, where they can then simply
90    paste the whole content of one of the original data tabs for the new spreadsheet to become fully functional.





| | A | B | C | D | E | F | G | H | I | J | K | L | M | N | O | P | Q |
|---|---|---|---|---|---|---|---|---|---|---|---|---|---|---|---|---|---|
| 1 | Longitude | 162.5 | degrees | | | Replace Pressure, Pratical Salinity (or Conductivity) and Temperature data (either ITS-90 or IPTS-68), including **Longitude** and **Latitude** | | | | | | | | | | | |
| 2 | Latitude | 33 | degrees | | | All colour colums will update automatically. DATA CAN BE DELETED BUT **NOT MOVED** prior to deletion OR THE FORMULAS WILL LOOSE THEIR REFERENCE. | | | | | | | | | | | |
| 3 | Conductivity (C) | Pressure (p) | Practical Salinity ($S_P$) | Temperature (t) ITS-90 | Temperature (t) IPTS-68 | $S_P$ from C | Reference Salinity ($S_R$) | delta $S_A$ Atlas | SAAR Atlas | Salinity Anomaly ($\delta S_A$) | Absolute Salinity ($S_A$) | t ITS-90 | Potential t ($\theta$) | Conservative Temperature ($\Theta$) | Potential Density ($\sigma_\Theta$) | In situ Density ($\rho_{S_A, \Theta, p}$) | Sound Speed (c) |
| 4 | (mS cm⁻¹) | (dbar) | | (ºC) | (ºC) | | (g kg⁻¹) | (g kg⁻¹) | (g kg⁻¹) | (g kg⁻¹) | (g kg⁻¹) | (ºC) | (ºC) | (ºC) | (kg m⁻³ - 1000) | (kg m⁻³) | (m s⁻¹) |
| 5 | | 0 | 34.57586 | 19.507610 | | 34.5759 | 34.7389 | 0.000327101505 | 0.000009410247 | 0.000326901616 | 34.7392 | 19.5076 | 19.5076 | 19.5130 | 24.5709 | 1024.5709 | 1519.5537 |
| 6 | | 10 | 34.74774 | 20.008300 | | 34.7477 | 34.9116 | 0.000339231758 | 0.000009777386 | 0.000341204433 | 34.9119 | 20.0083 | 20.0065 | 20.0072 | 24.5716 | 1024.6148 | 1521.2985 |
| 7 | | 20 | 34.67881 | 19.133780 | | 34.6788 | 34.8423 | 0.000333521900 | 0.000009747636 | 0.000339630409 | 34.8427 | 19.1338 | 19.1302 | 19.1319 | 24.7466 | 1024.8333 | 1518.9494 |
| 8 | | 30 | 34.68279 | 18.834320 | | 34.6828 | 34.8463 | 0.000375042687 | 0.000010581871 | 0.000368739417 | 34.8467 | 18.8343 | 18.8290 | 18.8302 | 24.8264 | 1024.9566 | 1518.2704 |
| 9 | | 40 | 34.68397 | 18.288160 | | 34.6840 | 34.8475 | 0.000389800378 | 0.000011376763 | 0.000396451974 | 34.8479 | 18.2882 | 18.2813 | 18.2817 | 24.9648 | 1025.1387 | 1516.8712 |
| 10 | | 50 | 34.68861 | 17.893830 | | 34.6886 | 34.8522 | 0.000430311850 | 0.000012931657 | 0.000450696471 | 34.8526 | 17.8938 | 17.8853 | 17.8852 | 25.0661 | 1025.2838 | 1515.8962 |
| 11 | | 76 | 34.69963 | 17.056150 | | 34.6996 | 34.8633 | 0.000569195077 | 0.000016219010 | 0.000565447460 | 34.8638 | 17.0561 | 17.0436 | 17.0423 | 25.2778 | 1025.6097 | 1513.8627 |
| 12 | | 101 | 34.69791 | 16.492310 | | 34.6979 | 34.8615 | 0.000696512528 | 0.000019779266 | 0.000689535385 | 34.8622 | 16.4923 | 16.4761 | 16.4742 | 25.4100 | 1025.8520 | 1512.5741 |
| 13 | | 126 | 34.71489 | 16.128460 | | 34.7149 | 34.8786 | 0.000843341842 | 0.000023960698 | 0.000835715273 | 34.8794 | 16.1285 | 16.1085 | 16.1059 | 25.5080 | 1026.0600 | 1511.8947 |
| 14 | | 151 | 34.68967 | 15.684310 | | 34.6897 | 34.8532 | 0.001001694449 | 0.000028396145 | 0.000989697859 | 34.8542 | 15.6843 | 15.6608 | 15.6585 | 25.5905 | 1026.2530 | 1510.9066 |
| 15 | | 176 | 34.65537 | 15.247770 | | 34.6554 | 34.8188 | 0.001146533554 | 0.000032958859 | 0.001147587425 | 34.8199 | 15.2478 | 15.2209 | 15.2191 | 25.6623 | 1026.4358 | 1509.9149 |
| 16 | | 202 | 34.63723 | 15.028760 | | 34.6372 | 34.8006 | 0.001306176088 | 0.000038329968 | 0.001333904319 | 34.8019 | 15.0288 | 14.9982 | 14.9967 | 25.6975 | 1026.5858 | 1509.6304 |
| 17 | | 252 | 34.58649 | 14.440070 | | 34.5865 | 34.7496 | 0.001555687022 | 0.000045929491 | 0.001596030542 | 34.7512 | 14.4401 | 14.4029 | 14.4022 | 25.7872 | 1026.8977 | 1508.5164 |
| 18 | | 303 | 34.53391 | 13.762160 | | 34.5339 | 34.6968 | 0.001918317195 | 0.000056693585 | 0.001967083277 | 34.6987 | 13.7622 | 13.7189 | 13.7189 | 25.8905 | 1027.2291 | 1507.0947 |
| 19 | | 353 | 34.44696 | 12.587460 | | 34.4470 | 34.6094 | 0.002406799849 | 0.000074277399 | 0.002570695638 | 34.6120 | 12.5875 | 12.5399 | 12.5411 | 26.0606 | 1027.6275 | 1503.9112 |
| 20 | | 404 | 34.37410 | 11.610510 | | 34.3741 | 34.5362 | 0.003092026444 | 0.000101116678 | 0.003492184663 | 34.5397 | 11.6105 | 11.5588 | 11.5609 | 26.1915 | 1027.9920 | 1501.3222 |
| 21 | | 505 | 34.17681 | 8.998112 | | 34.1768 | 34.3380 | 0.004943227283 | 0.000152354201 | 0.005231533736 | 34.3432 | 8.9981 | 8.9428 | 8.9471 | 26.4886 | 1028.7658 | 1493.3821 |
| 22 | | 606 | 34.04839 | 6.567234 | | 34.0484 | 34.2089 | 0.007328918773 | 0.000217880674 | 0.007453467487 | 34.2164 | 6.5672 | 6.5115 | 6.5167 | 26.7417 | 1029.5063 | 1485.5902 |
| 23 | | 707 | 34.05378 | 5.180429 | | 34.0538 | 34.2144 | 0.010134516840 | 0.000293431730 | 0.010039578297 | 34.2244 | 5.1804 | 5.1224 | 5.1270 | 26.9196 | 1030.1657 | 1481.6929 |
| 24 | | 808 | 34.13533 | 4.453866 | | 34.1353 | 34.2963 | 0.012801564089 | 0.000365710401 | 0.012542510966 | 34.3088 | 4.4539 | 4.3914 | 4.3949 | 27.0677 | 1030.7880 | 1480.4628 |
| 25 | | 909 | 34.21526 | 4.010992 | | 34.2153 | 34.3766 | 0.014899810655 | 0.000425738110 | 0.014635428541 | 34.3912 | 4.0110 | 3.9430 | 3.9457 | 27.1798 | 1031.3704 | 1480.3736 |
| 26 | | 1010 | 34.28701 | 3.630195 | | 34.2870 | 34.4487 | 0.016512321555 | 0.000473242399 | 0.016302579738 | 34.4650 | 3.6302 | 3.5568 | 3.5588 | 27.2768 | 1031.9375 | 1480.5196 |
| 27 | | 1111 | 34.33858 | 3.351280 | | 34.3386 | 34.5005 | 0.017685037469 | 0.000508934903 | 0.017558509209 | 34.5181 | 3.3513 | 3.2720 | 3.2736 | 27.3463 | 1032.4752 | 1481.0620 |
| 28 | | 1213 | 34.38449 | 3.102174 | | 34.3845 | 34.5466 | 0.018687486637 | 0.000538076023 | 0.018588711984 | 34.5652 | 3.1022 | 3.0169 | 3.0181 | 27.4074 | 1033.0086 | 1481.7362 |
| 29 | | 1314 | 34.42426 | 2.876307 | | 34.4243 | 34.5866 | 0.019494191089 | 0.000561454629 | 0.019418798328 | 34.6060 | 2.8763 | 2.7851 | 2.7861 | 27.4607 | 1033.5292 | 1482.4828 |
| 30 | | 1416 | 34.45672 | 2.694073 | | 34.4567 | 34.6192 | 0.020176311159 | 0.000582091514 | 0.020151541483 | 34.6393 | 2.6941 | 2.5966 | 2.5974 | 27.5037 | 1034.0426 | 1483.4231 |
| 31 | | 1517 | 34.48842 | 2.506860 | | 34.4884 | 34.6510 | 0.020753899596 | 0.000599223978 | 0.020763738622 | 34.6718 | 2.5069 | 2.4034 | 2.4040 | 27.5459 | 1034.5510 | 1484.3241 |
| 32 | | 1771 | 34.54501 | 2.196994 | | 34.5450 | 34.7079 | 0.021602210963 | 0.000617441618 | 0.021430104714 | 34.7293 | 2.1970 | 2.0767 | 2.0772 | 27.6186 | 1035.7882 | 1487.2625 |
| 33 | | 2025 | 34.58881 | 1.953304 | | 34.5888 | 34.7519 | 0.021906208811 | 0.000625378399 | 0.021733094502 | 34.7736 | 1.9533 | 1.8154 | 1.8157 | 27.6744 | 1037.0030 | 1490.4856 |
| 34 | | 2279 | 34.61341 | 1.825799 | | 34.6134 | 34.7766 | 0.021834954336 | 0.000625665801 | 0.021758546231 | 34.7984 | 1.8258 | 1.6682 | 1.6684 | 27.7053 | 1038.1828 | 1494.2095 |
| 35 | | 2534 | 34.63526 | 1.709283 | | 34.6353 | 34.7986 | 0.021631273849 | 0.000621030221 | 0.021610969888 | 34.8202 | 1.7093 | 1.5312 | 1.5314 | 27.7329 | 1039.3583 | 1498.0128 |
| 36 | | 2789 | 34.64812 | 1.626311 | | 34.6481 | 34.8115 | 0.021385378943 | 0.000614436829 | 0.021389468131 | 34.8329 | 1.6263 | 1.4264 | 1.4267 | 27.7507 | 1040.5169 | 1501.9703 |
| 37 | | 3045 | 34.65890 | 1.566066 | | 34.6589 | 34.8223 | 0.021139155635 | 0.000607589802 | 0.021157693528 | 34.8435 | 1.5661 | 1.3432 | 1.3434 | 27.7651 | 1041.6695 | 1506.0602 |
| 38 | | 3300 | 34.66710 | 1.527974 | | 34.6671 | 34.8306 | 0.020918230176 | 0.000601218243 | 0.020940774256 | 34.8515 | 1.5280 | 1.2809 | 1.2811 | 27.7759 | 1042.8067 | 1510.2450 |
| 39 | | 3556 | 34.67315 | 1.503932 | | 34.6732 | 34.8366 | 0.020701479725 | 0.000595069372 | 0.020730222690 | 34.8574 | 1.5039 | 1.2314 | 1.2317 | 27.7841 | 1043.9387 | 1514.5230 |
| 40 | | 3812 | 34.67689 | 1.489079 | | 34.6769 | 34.8404 | 0.020516621253 | 0.000589541645 | 0.020539870502 | 34.8609 | 1.4891 | 1.1901 | 1.1903 | 27.7898 | 1045.0614 | 1518.8548 |

TEOS-10 Test Data | TS-55 | CTD-020 | Surface Data | Vertical Profiles | SA - Θ Diagram | longs_ref | lats_ref | ndepth_ref | p_ref | deltaSA_ref | SAAR_ref

**Figure 1:** TEOS-10 EXCEL workbook green data tab. Seawater properties in coloured columns are computed on the fly from user data pasted into white cells.

## 2.1.1 Data input

- Location: The green tab's data template was developed to process vertical casts located at a given location. *Longitude* and *latitude* must be input in cells 'B1:B2' in decimal format (degrees). *Longitude* can either be within the domain (-180º to 180º) or (0º to 360º) i.e., 10º 30' W can either be input as -10.5º or 349.5º. The *latitude* domain is (-90º to 90º) i.e., 30º S would be -30º. The input of the cast coordinates is essential, as *absolute salinity* is dependent of location (Sect 3.6).

- Pressure: *pressure* (*p*) units are dbar. For the upper ocean, 10 dbar ~ 10 m.

- Salinity: the salinity quantity is *practical salinity* ($S_P$) which continues to be the recommended salinity quantity to be archived (IOC, SCOR and IAPSO, 2010). *Practical salinity* ($S_P$) is obtained from conductivity and the EOS-80 polynomials for estimating $S_P$ still apply. Oceanographic instruments that measure in-situ conductivity, output their measurements usually in *conductivity – temperature - pressure* triplets and so *conductivity* (mS cm⁻¹) might be archived instead of $S_P$. If this is the case, *conductivity* may be used as input data (column 'A') instead of *practical salinity*. Column 'F' of the spreadsheet ('$S_P$ from C') checks if there are $S_P$ data in column 'C' and if not, computes $S_P$ from the conductivity data using the {*SP_from_C(C, t, p)*} function (note: EOS-80 polynomials use temperature IPTS-68 as argument, while TEOS-10 functions expect temperature to be ITS-90; for consistency, the temperature argument for this function is ITS-90 and the first line of code converts temperature back to IPTS-68).





- Temperature: t*emperature* (ºC) should be input in column 'D' or 'E', respectively if it is ITS-90 or IPTS-68 (data sets before 1990 are in the IPTS-68 standard, but recent data can still be using this standard instead of the newer ITS-90 – checking the instrument specifications and/or the metadata associated with the data is advisable). Column 'L' of the spreadsheet ('t ITS-90') either uses data in column 'D', if it exists, or converts the ITS-68 values to ITS-90 (ITS-

90 = ITS-68 / 1.00024). All functions use temperature ITS-90 as input.

## 2.2 The yellow 'Surface data' tab

The yellow 'Surface data' tab content differs from the other data spreadsheets on what refers to the input of the location coordinates. In this spreadsheet, longitude and latitude are input in the first two columns, allowing in this way the assignment of distinct coordinates for each line. This is useful if the data set is not a vertical cast at a given location but a set of

measurements on different locations, typically at the same pressure level (e.g., surface measurements). The data included are 'fictitious' and used here to demonstrate the use of the template. The location of the first four data lines is in the Baltic Sea. Conditions in the Baltic are different from the open ocean (McDougall, 2010) and TEOS-10 treats this adjacent sea as being a specific case – while for the world ocean the estimation of $S_A$ depends on the measured salinity anomaly values at that location (look-up tables), for the Baltic it is estimated by Eq. (1).


$$S_A\ (Baltic) = \left(\frac{35.16504 - 0.087}{35}\right) \times\ S_P + 0.087 \qquad (1)$$

Whenever the location is in the Baltic (which is checked by the {*is_Baltic(lon, lat)*} function), the salinity anomaly cells display 'Baltic'. This spreadsheet also includes a line with data from line one of the 'TEOS-10 Test Data' tab (surface data

from the NW Pacific) as well as a location over land, which returns 'NOT in OCEAN' for the look-up table cells, and an error for the other parameters.

| | Long | Lat | Conductivity (C) | Pressure (p) | Practical Salinity (S_P) | Temperature (t) ITS-90 | Temperature (t) IPTS-68 | S_P from C | Reference Salinity (S_R) | delta S_A Atlas | SAAR Atlas | Salinity Anomaly (δS_A) | Absolute Salinity (S_A) | t ITS-90 | Potential t (θ) | Conservative Temperature (Θ) | Potential Density (σ_θ) | In situ Density (ρ_SA, θ, p) | Sound Speed (c) |
|---|---|---|---|---|---|---|---|---|---|---|---|---|---|---|---|---|---|---|---|
| | degrees | degrees | (mS cm⁻¹) | (dbar) | | (°C) | (°C) | | (g kg⁻¹) | (g kg⁻¹) | (g kg⁻¹) | (g kg⁻¹) | (g kg⁻¹) | (°C) | (°C) | (°C) | (kg m⁻³ - 1000) | (kg m⁻³) | (m s⁻¹) |
| 5 | 20.05 | 59.02 | | 0 | 5.39 | 12.3000 | | 5.3900 | 5.4154 | Baltic | Baltic | Baltic | 5.4890 | 12.3000 | 12.3000 | 12.8682 | 3.7047 | 1003.7047 | 1462.7763 |
| 6 | 20.1 | 59.02 | | 0 | 5.39 | 12.2000 | | 5.3900 | 5.4154 | Baltic | Baltic | Baltic | 5.4890 | 12.2000 | 12.2000 | 12.7641 | 3.7177 | 1003.7177 | 1462.4072 |
| 7 | 20.15 | 59.02 | | 0 | 5.38 | 12.1000 | | 5.3800 | 5.4054 | Baltic | Baltic | Baltic | 5.4790 | 12.1000 | 12.1000 | 12.6601 | 3.7228 | 1003.7228 | 1462.0253 |
| 8 | 20.2 | 59.02 | | 0 | 5.41 | 12.1000 | | 5.4100 | 5.4355 | Baltic | Baltic | Baltic | 5.5091 | 12.1000 | 12.1000 | 12.6596 | 3.7460 | 1003.7460 | 1462.0613 |
| 9 | | | | | | | | | | 0.000575919565 | 0.00001605931B | | | | | | | | |
| 10 | 162.5 | 33 | | 0 | 34.57586 | 19.507610 | | 34.5759 | 34.7389 | 0.000327101505 | 0.000009410247 | 0.000326901616 | 34.7392 | 19.5076 | 19.5076 | 19.5130 | 24.5709 | 1024.5709 | 1519.5537 |
| 11 | | | | | | | | | | 0.000575919565 | 0.00001605931B | | | | | | | | |
| 12 | 2 | 11 | | 0 | 34.57586 | 19.507610 | | 34.5759 | 34.7389 | NOT in OCEAN | NOT in OCEAN | #VALUE! | #VALUE! | 19.5076 | #VALUE! | #VALUE! | #VALUE! | #VALUE! | #VALUE! |

Row 1: Replace **Pressure**, **Pratical Salinity** (or **Conductivity**) and **Temperature** data (either ITS-90 or IPTS-68), including **Longitude** and **Latitude**

Row 2: All colour colums will update automatically. DATA CAN BE DELETED BUT **NOT MOVED** prior to deletion OR THE FORMULAS WILL LOOSE THEIR REFERENCE.

Tabs: TEOS-10 Test Data | TS-55 | CTD-020 | Surface Data | Vertical Profiles | SA - Θ Diagram | longs_ref | lats_ref | ndepth_ref | p_ref | deltaSA_ref | SAAR_ref

**Figure 2:** TEOS-10 EXCEL workbook 'Surface data' tab. Surface data from different locations (location coordinates for each line). Four

samples are from the Baltic Sea, one from the NW Pacific and one location is over land.



## 2.3 Vertical Profiles tab

The 'Vertical profiles' tab includes five plots that use the 'TS-55' and 'CTD-020' data sets and one plot with the 'TEOS-10 Test Data' data set. Two of these plots are reproduced in Figs. 3 and 4. Changing the data will update the plots accordingly, and the user can add extra profiles by right clicking the plot area, click 'Select Data' and edit/update the data sources.

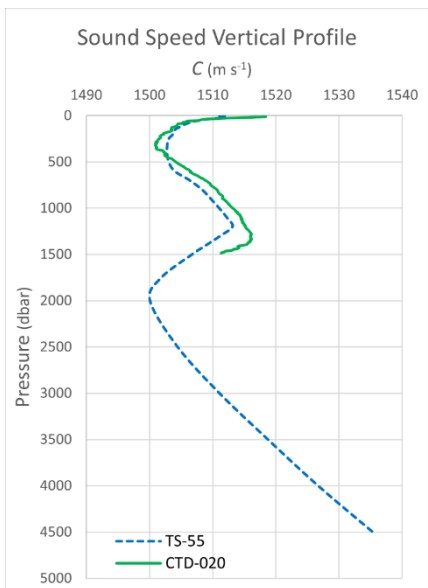

**Figure 3:** Sound speed vertical profile of two data sets included in TEOS-10 EXCEL. This plot is one of six included in the 'Vertical Profiles' tab.

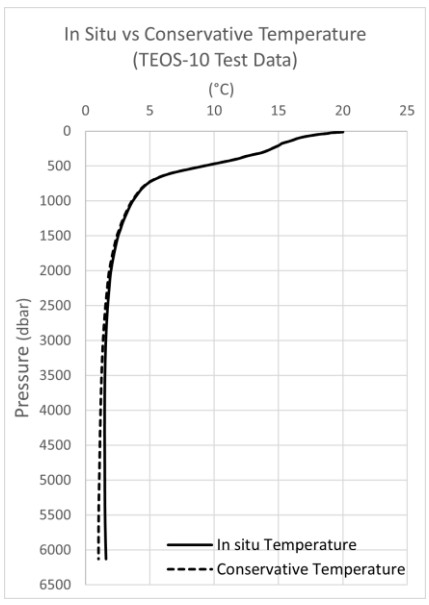



**Figure 4:** Comparison between *in situ* and *conservative temperature* of the TEOS-10 Test Data included in TEOS-10 EXCEL. This plot is one of six included in the 'Vertical Profiles' tab.

### 2.4 S$_A$ – Θ Diagram tab

This is a template for plotting *absolute salinity* (S$_A$) – *conservative temperature* (Θ) diagrams. Since the introduction of TEOS-10, S$_A$ – Θ diagrams have replaced T-S diagrams (EOS-80) for the characterisation of water masses in the ocean. The two diagrams represented are from the NW Pacific ('TEOS-10 Test Data') and NE Atlantic ('TS-55'). Users can right-click the plot area, click 'Select Data' and edit/update the data sources. The plot also shows the pressure values at selected points along the two S$_A$ – Θ diagram lines (the label of the points is retrieved from the pressure data column in the respective data tab).

These points can be individually selected and edited. The density field is shown through a set of $\sigma_\Theta$ dashed lines obtained from S$_A$ – Θ pairs that resolve to constant values of $\sigma_\Theta$ (i.e., 24, 25, …, 29). As in all data spreadsheets, white cells can be edited. In this case, S$_A$ spans from the *x*-axis minimum (33.0) to the *x*-axis maximum (38.0) with a 0.05 increment. The *conservative temperature* (Θ) values (green columns) are obtained by the function {*sigma_CT_line(SA, sigma, min_temp, max_temp*}. If more $\sigma_\Theta$ lines are desired, additional column pairs can be inserted into the spreadsheet and new series added to the plot.






**Figure 5:** *Absolute salinity* ($S_A$) *– conservative temperature* (Θ) *diagrams*. Data are from the 'TEOS-10 Test Data' (NW Pacific) and 'TS-55' tab (NE Atlantic off the Iberian Peninsula). Pressure values are obtained dynamically from the data and the density field ($\sigma_\Theta$) is computed from data included in the spreadsheet template.

## 2.5 The TEOS-10 Look-up tabs (purple) *aka* the 'Atlas'

Figure 6 shows the [ndepth_ref] look-up table which contains the number of pressure levels in each of the seawater samples that constitute the 'Atlas'. If looking closer to this spreadsheet, it becomes apparent that the empty cells represent land, and the 'white' shapes approximate to a map of the world land masses. The top of the spreadsheet is the South Pole, bottom is the North Pole, and the Greenwich Meridian (0º longitude) is at the left. *Longitude* is positive to the right, so actually the 'map' is a skewed mirror image of the earth surface. Nonetheless, mirrored continental shapes are identifiable. The *longitude* bins (or cells) are referenced in the [longs_ref] look-up table, so the *longitude* of the cell highlighted in green at the upper left of Fig. 6, which *x*-coordinate is 4 (fourth column), is 12º East (value at the fourth line of the [longs_ref] table). This cell is at line nine (*y*-coordinate = 9) of the [ndepth_ref] table (Fig. 6) which, looking up in the [lats_ref] table, corresponds to -54º of *latitude*. The green cell in Fig. 1, corresponds though to a reference cast located at 54º S, 12º E, with 41 pressure levels. These 41




pressure levels correspond to the *pressure* values indicated in the [p_ref] table. The [p_ref] table has level 22 (1010 dbar) highlighted in green, as this 3D location is used and referenced as a debug point in the {*LookUp_atlas(table_name, p, lon, lat)*} function, that retrieves data from the *absolute salinity anomaly* [deltaSA_ref] and the *absolute salinity anomaly ratio*

[SAAR_ref] look-up tables. The [ndepth_ref] table (Fig. 6) has 45 lines by 91 columns and so [deltaSA_ref] and [SAAR_ref] which have both the same size, have 4,095 columns (45 x 91) by 45 lines (pressure levels). An additional numbering line (which does not affect how data is located) was added to facilitate debugging. Position (column number) in these tables is given by Eq. (2).

$$Column = (x-1) * nlat + y \qquad (2)$$

In Eq. (2), *x* is the longitude bin, *y* the latitude bin and *nlat* the number of latitude bins (45). For the 'green' cell (Fig. 1), *x=4* and *y=9* as mentioned before, so the anomaly data for this cell is located at column 144, line 22 (which corresponds to 1010 dbar) of the [deltaSA_ref] table. The reference *salinity anomaly* at 54º S, 12º E, 1010 dbar, is 0.008323162 g kg$^{-1}$ (also

highlighted in green).

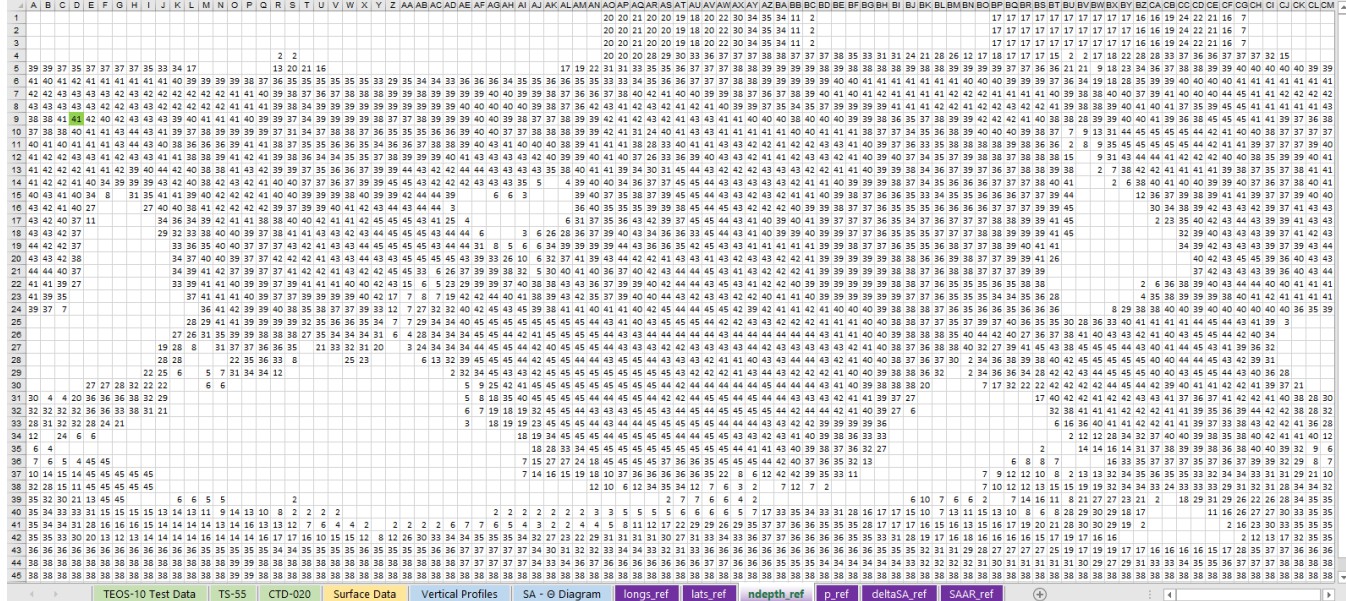

**Figure 6:** [ndepth_ref] look-up table. The table has 45 rows (latitude) by 91 columns (longitude). South is at the top (1$^{st}$ row is 86º S) and 1$^{st}$ column is 0º of longitude. The latitude x longitude grid is a 4º x 4º grid and each cell location is obtained from the [longs_ref] and [lats_ref]
tables. Cell values are the number of *pressure* levels at the given location. The cell highlighted in green is used as a 'case study' in the text.





### 3. VBA (Visual Basic for Applications) functions

Most functions of TEOS-10 EXCEL are a direct translation into VBA of the GSW MATLAB counterpart (McDougall and Barker, 2011) and the original credit and references were kept in the code comments. However, due to the different way matrices are handled in MATLAB versus VBA, some functions needed to be utterly redesigned, namely on what concerns
accessing the 'Atlas' look-up tables. Returned values from TEOS-10 EXCEL are the same, for every parameter, as the ones obtained with the GSW toolbox, up to 15 decimal places, i.e., difference = 0.000000000000000 (error checking was performed against MATLAB GSW Toolbox version 3.06.12 from 25[th] May 2020). As referred before, access to the VBA project environment is obtained by pressing [Alt – F11]. All functions are described next, following the spreadsheet's column sequence.

### 3.1. $S_P$ from $C$

Function {$SP\_from\_C$ $(C, t, p)$} computes *practical salinity* ($S_P$) from conductivity using the EOS-80 Fofonoff and Millard (1983) equations.  For consistency with all TEOS-10 functions, the temperature argument is ITS-90 (the first line of code converts temperature back to IPTS-68 as expected in EOS-80). *Practical salinity* is a dimensionless quantity, although PSU (Practical Salinity Units) is commonly used. For reference, the conductivity of Standard Sea Water at $S_P = 35$, $t_{68} = 15$, $p = 0$
is 42.9140 mS cm$^{-1}$, which can be used to validate the function.

### 3.2 Reference Salinity ($S_R$)

*Reference salinity* ($S_R$) is assumed to be proportional to *practical salinity* (IOC, SCOR and IAPSO, 2010) and obtained by Eq. (3). Units for $S_R$ are g kg$^{-1}$.

$$S_R = \frac{35.16504}{35} \times S_P \hspace{3cm} (3)$$

### 3.3 delta $S_A$ Atlas

Function {$LookUP\_atlas(table\_name, p, lon, lat)$} interrogates the 'Atlas' database and was developed specifically for TEOS-10 EXCEL. The argument '*table_name*' can be one of the two look-up tables ("deltaSA_ref" or "SAAR_ref") and the returned values are a 3D interpolation of the 8 vertices of the cube around the location (Fig. 7). For the [deltaSA_ref] table, the result of the function is the atlas *absolute salinity anomaly* ($\delta S_A{}^{atlas}$). As the interpolation process is not very clearly described in the
GSW toolbox documentation, it is discussed next.

#### 3.3.1 Interpolation

Function {$LookUP\_atlas(table\_name, p, lon, lat)$} begins by finding the grid point P1 of the 3D cube around the location (Fig. 7). P1 would be the grid point immediately before the latitude and longitude of a given location. The same applies to pressure. For example, if the spreadsheet data cell corresponds to a cast located at 1100 dbar, +13° longitude, -51° latitude, the grid


position of P1(lon*, lat*, p*) would be P1(4, 9, 22), obtained from the [longs_ref], [lats_ref] and [p_ref] tables. The other 8

points are referenced to P1, by adding one unit to the grid position of P1 as shown in Fig. 1.

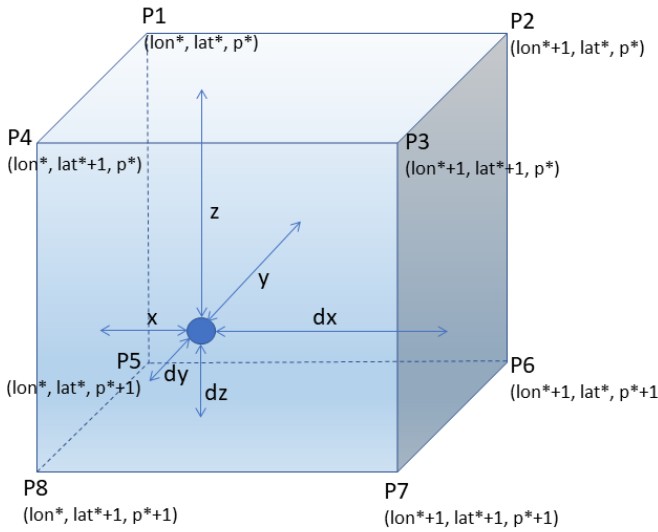

**Figure 7:** 3D interpolation cube. Points are defined by their grid position (lon*, lat*, p*).

The standard basic 3D interpolation model assumes that the cube dimensions are 1 x 1 x 1 (Bourke, 1999) and the distances

$dx$, $dy$ and $dz$ are obtained by subtracting, respectively, $x$, $y$, and $z$, from the unit. However, in this case, the longitude and

latitude grid space are 4° and the pressure difference between the upper and bottom points varies from grid level to grid level

(e.g., 10 dbar between levels 1 and 2 but 101 dbar between levels 22 and 23). Distance $x$, $y$ and $z$ are obtained from Eqs. (4, 5

and 6), and then $dx$, $dy$ and $dz$ by Eq. (7)


$$x = (lon - lon(P1))/4 \qquad\qquad (4)$$
$$y = (lat - lat(P1))/4 \qquad\qquad (5)$$
$$z = (p - p(P1))/(p(P5) - p(P1)) \qquad\qquad (6)$$
$$dx = 1 - x, \; dy = 1 - y, \; dz = 1 - z \qquad\qquad (7)$$


The interpolated value ($v$) is obtained by weighing the contribution of the eight points according to Eqs. (8 to 16), where $v(Pn)$

is the $\delta S_A{}^{\text{atlas}}$ value at P$n$ (from [deltaSA_table]).

$$v1 = v(P1) \times dx \times dy \times dz \qquad\qquad (8)$$
$$v2 = v(P2) \times x \times dy \times dz \qquad\qquad (9)$$
$$v3 = v(P3) \times x \times y \times dz \qquad\qquad (10)$$




$$v4 = v(P4) \times dx \times y \times dz \tag{11}$$

$$v5 = v(P5) \times dx \times dy \times z \tag{12}$$

$$v6 = v(P6) \times x \times dy \times z \tag{13}$$

$$v7 = v(P7) \times x \times y \times z \tag{14}$$

$$v8 = v(P8) \times dx \times y \times z \tag{15}$$

$$v = v1 + v2 + v3 + v4 + v5 + v6 + v7 + v8 \tag{16}$$

### 3.3.2 Missing data

There are pressure levels in the atlas reference casts where data is missing. Figure 8 illustrates this situation.


| | G | H | I | J | K | L | M | N | O | P | Q | R |
|---|---|---|---|---|---|---|---|---|---|---|---|---|
| 23 | 0.009218692 | 0.008908996 | 0.00759247 | 0.005709105 | 0.004574747 | 0.003644211 | 0.003208049 | 0.003031175 | 0.00349246 | 0.003986676 | 0.003968637 | 0.004115448 |
| 24 | 0.009233077 | 0.008972897 | 0.007761753 | 0.005780614 | 0.004753488 | 0.004032915 | 0.003684931 | 0.003560448 | 0.003808101 | 0.003997765 | 0.003876938 | 0.003981588 |
| 25 | 0.009249478 | 0.009033516 | 0.007911187 | 0.005844102 | 0.004886662 | 0.004367845 | 0.004065591 | 0.00396472 | 0.003978199 | 0.003921243 | 0.003732477 | 0.003787497 |
| 26 | 0.00925642 | 0.009087358 | 0.008098981 | 0.005932269 | 0.00497313 | 0.004602549 | 0.004352101 | 0.004216718 | 0.004038171 | 0.003810867 | 0.003586259 | 0.00359505 |
| 27 | 0.009258621 | 0.009120991 | 0.008237284 | 0.006021737 | 0.005026371 | 0.004782731 | 0.00458536 | 0.004400239 | 0.00401516 | 0.003677485 | 0.003438529 | 0.003469429 |
| 28 | 0.009248313 | 0.009169287 | 0.008541587 | 0.006386436 | 0.005105027 | 0.004895867 | 0.004721168 | 0.004429583 | 0.003842076 | 0.003487118 | 0.003261295 | 0.003335104 |
| 29 | 0.009245437 | 0.009196107 | 0.008768696 | 0.006796146 | 0.005237766 | 0.004803792 | 0.004612022 | 0.004309108 | 0.003815922 | 0.003634408 | 0.003400565 | 0.003466424 |
| 30 | 0.009226827 | 0.009208929 | 0.008945121 | 0.007129832 | 0.005488165 | 0.004770552 | 0.004442442 | 0.004210304 | 0.003888642 | 0.003889622 | 0.003755912 | 0.0038033 |
| 31 | 0.009197872 | 0.009206341 | 0.009035997 | 0.007610069 | 0.005890486 | 0.004872129 | 0.004405444 | 0.004180367 | 0.00394193 | 0.004112428 | 0.004104743 | 0.004197571 |
| 32 | 0.009170239 | 0.009187315 | 0.009141968 | 0.007932154 | 0.006315794 | 0.005045773 | 0.004471019 | 0.004192354 | 0.003957406 | 0.00424212 | 0.004378315 | 0.004611299 |
| 33 | 0.00913718 | | | | | 0.005321122 | 0.004571465 | 0.004236636 | 0.003977235 | 0.004339081 | 0.004612237 | 0.004979521 |
| 34 | 0.009109314 | | | | | 0.005831923 | 0.004945996 | 0.004390707 | 0.004032286 | 0.004419669 | 0.004770676 | 0.005217486 |
| 35 | 0.00908662 | 0.009045589 | 0.00913435 | 0.008461061 | 0.00765709 | 0.00638616 | 0.005509364 | 0.004780258 | 0.004146816 | 0.004491119 | 0.004894749 | 0.005350943 |
| 36 | 0.009041758 | 0.008978659 | 0.009089345 | 0.008494994 | 0.007976087 | 0.007045019 | 0.006425593 | 0.005681561 | 0.004481928 | 0.004559633 | 0.004962076 | 0.005420412 |
| 37 | 0.008951321 | 0.008872803 | 0.009039996 | 0.008480566 | 0.008165775 | 0.007627993 | 0.007342053 | 0.006830766 | 0.005038608 | 0.004740746 | 0.005007445 | 0.005467756 |
| 38 | 0.008820467 | 0.008753362 | 0.009175537 | | 0.008282431 | 0.007790812 | 0.007626658 | 0.007408762 | 0.005235964 | 0.004810099 | 0.005035742 | 0.005505476 |
| 39 | 0.008768482 | 0.00867393 | | | 0.008292218 | 0.007935395 | 0.007805511 | 0.007774958 | 0.005151711 | 0.00504291 | 0.005070624 | 0.005536356 |
| 40 | 0.008739336 | 0.008706534 | | | 0.008598181 | 0.007909775 | 0.007863286 | 0.007845305 | 0.00539689 | 0.005264025 | 0.005140008 | 0.005569826 |
| 41 | 0.008702934 | 0.008686641 | | | | 0.007908649 | 0.007888701 | 0.007852586 | | 0.005455169 | 0.005213869 | 0.005591903 |
| 42 | 0.008659186 | 0.008640046 | | | | | | | | 0.004816461 | 0.00509936 | 0.005612582 |
| 43 | | 0.008632253 | | | | | | | | 0.00469658 | 0.005234804 | 0.005692186 |
| 44 | | | | | | | | | | | | |
| 45 | | | | | | | | | | | | |
| 46 | 7 | 8 | 9 | 10 | 11 | 12 | 13 | 14 | 15 | 16 | 17 | 18 |

| TEOS-10 Test Data | TS-55 | CTD-020 | Surface Data | Vertical Profiles | SA - Θ Diagram | longs_ref | lats_ref | ndepth_ref | p_ref | deltaSA_ref | SAAR … |

**Figure 8:** [deltaSA_ref] table: reference data missing for pressure levels 33 and 34 of columns 8, 9, 10 and 11.

The GSW toolbox fills these gaps by averaging the neighbouring four points in the grid at the same pressure level. As the
ocean is horizontally stratified, this makes 'oceanographic sense', but neighbour points, themselves, might also lack data at
the same level, which might compromise the result. A $\delta S_A{}^{atlas}$ plot from [deltaSA_ref] column 8 (0º lon, -54º lat) is shown in
Fig. 9. The $\delta S_A{}^{atlas}$ output of GSW toolbox and TEOS-10 EXCEL is the same for the part where there are data (blue line), but
the output of the GSW toolbox for the two pressure levels where data is missing is clearly off the profile (orange). In this
situation, TEOS-10 EXCEL implements a vertical interpolation within the vertical profile, which resolves better the missing
data situations (green in Fig.9). The second case where GSW toolbox seems to be less consistent, is when not all points of the
3D interpolation cube exist at the last pressure level. The GSW toolbox test data (included in the 'TEOS-10 Test Data' tab) is
such an example (Fig. 10). The GSW toolbox approach for resolving missing data on the last pressure level is as before. In
these situations, if one of the four bottom points of the interpolation cube (P5, P6, P7 or P8 in Fig. 7) is missing, TEOS-10

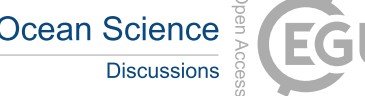

EXCEL assigns to it the same value of the next upper point at that location (e.g., $v(P7) = v(P3)$). Again, the resulting profile is

more coherent in TEOS-10 EXCEL than in the GSW toolbox (Fig. 10).

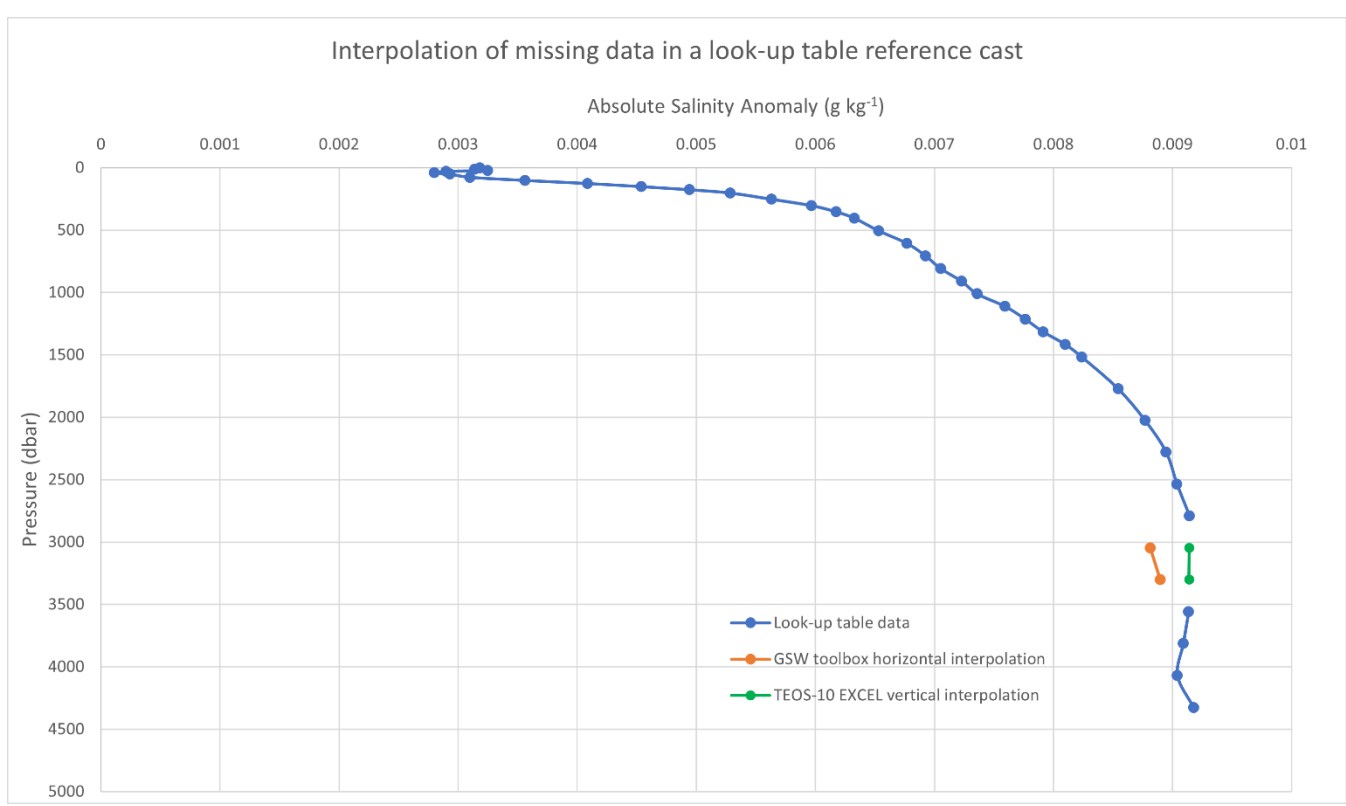

**Figure 9:** $\delta S_A{}^{atlas}$ plot from [deltaSA_ref] column 8. Missing data at levels 33 and 34 (3045 and 3300 dbar) is not well resolved by horizontal interpolation (GSW toolbox). Vertical interpolation implemented in TEOS-10 EXCEL resolves better these situations.


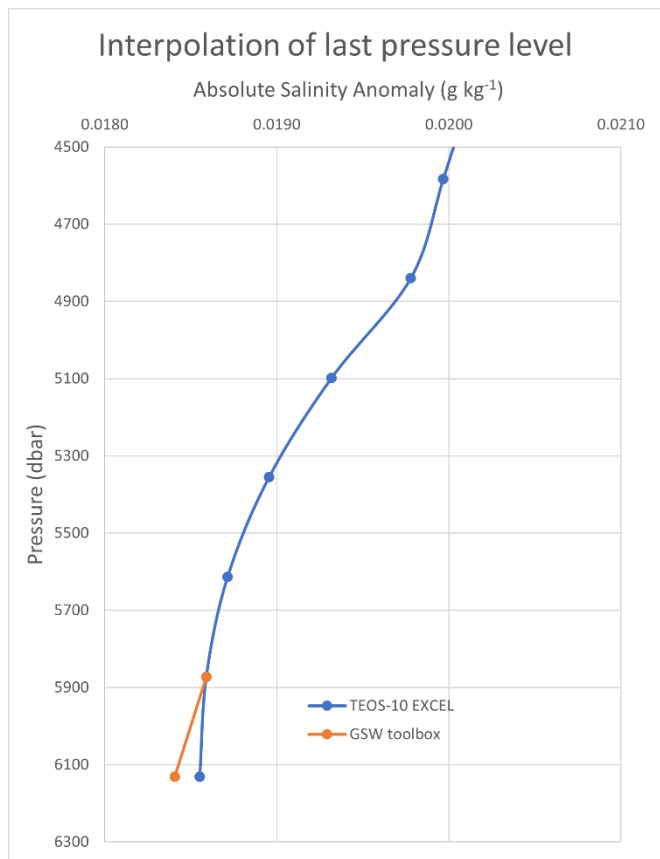


**Figure 10:** $\delta S_A{}^{\text{atlas}}$ plot from 'TEOS-10 Test Data' tab. Not all neighbour points have data at the last pressure level. For these points, TEOS-10 EXCEL uses the same value as of their last pressure level to resolve the 3D interpolation cube, while the GSW toolbox averages the same level data from neighbouring points.

**3.4 delta SAAR Atlas**

The atlas *absolute salinity anomaly ratio* ($R^\delta$) is obtained by the function {*LookUP_atlas(table_name, p, lon, lat)*}, exactly as for the atlas *absolute salinity anomaly* (Sect. 3.3), but calling it with "SAAR_ref" as the *table_name* argument. $R^\delta$ is the quantity used to estimate *absolute salinity anomaly* (Sect. 3.5) which is then used for obtaining *absolute salinity* (Sect. 3.6).

**3.5 Absolute Salinity Anomaly**

Absolute s*alinity anomaly* ($\delta S_A$) is the product of the atlas *absolute salinity anomaly ratio* by *reference salinity* (Eq. 17).

$\delta S_A = R^\delta \times S_R$              (17)

**3.6 Absolute Salinity**

*Absolute salinity* ($S_A$) is the sum of *reference salinity* and *salinity anomaly* (Eq. 18).

$S_A = S_R + \delta S_A$     (g kg$^{-1}$)        (18)





If the location is in the Baltic Sea, the world atlas salinity anomalies do not apply (McDougall, 2010) and *absolute salinity* is

computed algebraically from *practical salinity* with function {*SA_Baltic(SP)*}, which applies Eq. (19).

$$S_A^{Baltic} = \frac{(35.16504 - 0.087)}{35} \times S_P + 0.087 \quad \text{(g kg}^{-1}\text{)} \qquad (19)$$

Limits for the Baltic were taken from Fig. 2 of Feistel et al. (2010). Function {*is_Baltic(lon, lat)*} checks if the location is in

the Baltic Sea by finding if the coordinates lie within any of two rectangular areas defined by [9° E : 15° E; 52° N : 60° N] and

[15° E : 30° E; 52° N : 67° N].

### 3.7 Temperature ITS-90

The temperature standard used in TEOS-10 for temperature measurements is ITS-90 (Preston-Thomas, 1990). If column 'D'

of the data spreadsheet (temperature ITS-90) contains data, these values will be used, if the input temperature is IPTS-68

(column 'E'), it will be converted to ITS-90 using Eq. (20).

$$t_{ITS-90} = {t_{IPTS-68}}/{1.00024} \quad \text{(°C)} \qquad (20)$$

### 3.8 Potential Temperature ($\theta$)

*Potential temperature* ($\theta$, °C) is obtained by function {*pt0_from_t(SA,t,p)*}. Three other functions, {*Entropy_part(SA, t, p)*},

{*Entropy_part_zerop(SA, pt0)*}, {*Gibbs_pt0_pt0(SA, pt0)*} are called within the computation process. These four functions

are a VBA translation of the original GSW toolbox counterparts (IOC, SCOR and IAPSO, 2010; McDougall and Wotherspoon,

2013).

### 3.9 Conservative Temperature ($\Theta$)

*Conservative temperature* ($\Theta$, °C) is the temperature quantity used as argument in the Thermodynamic Equation Of Seawater

- 2010 (IOC, SCOR and IAPSO, 2010). It is obtained with function {*CT_from_pt(SA, pt)*} which is a direct translation from

the GSW toolbox counterpart and estimates $\Theta$ from $S_A$ and $\theta$.

### 3.10 Potential Density ($\sigma_\Theta$)

*Potential density* ($\sigma_\Theta$, kg m$^{-3}$ - 1000) with reference to sea pressure of 0 dbar is estimated by function {*sigma0(SA, CT)*}, which

arguments are *absolute salinity* ($S_A$) and *conservative temperature* ($\Theta$). This function uses the TEOS-10 75-term equation and

is a VBA translation of the original GSW toolbox counterpart (IOC, SCOR and IAPSO, 2010; McDougall et al., 2003; Roquet

et al., 2015).





### 3.11 In situ Density ($\rho_{S_A, \Theta, p}$)

*In situ density* ($\rho_{S_A, \Theta, p}$, kg m$^{-3}$) is estimated by function {*rho(SA, CT, p)*}, which arguments are *absolute salinity* ($S_A$), *conservative temperature* ($\Theta$) and *pressure* ($p$). This function uses the TEOS-10 75-term equation and is a VBA translation of

the original GSW toolbox counterpart (IOC, SCOR and IAPSO, 2010; McDougall et al., 2003; Roquet et al., 2015).

### 3.12 Sound speed (*c*)

*Sound speed* (*c*, m s$^{-1}$) is estimated by function {*Sound_Speed(SA, CT, p)*}, which arguments are *absolute salinity* ($S_A$), *conservative temperature* ($\Theta$) and *pressure* ($p$). This function uses the TEOS-10 75-term equation and is a VBA translation of the original GSW toolbox counterpart (IOC, SCOR and IAPSO, 2010; McDougall et al., 2003; Roquet et al., 2015).

### 3.13 [deltaSA_ref] table not needed!

The atlas *absolute salinity anomaly* (column 'H' of the data spreadsheets) is not used for any calculation, as *absolute salinity anomaly* ($\delta S_A$) is obtained from the product of *absolute salinity anomaly ratio* ($R^\delta$) by *reference salinity* (Eq. 17), and $R^\delta$ is retrieved from the [SAAR_ref] look-up table. The [deltaSA_ref] look-up table is though not necessary for any computation and can eventually be deleted from the EXCEL workbook to lighten up the code. However, this table was used as a debugging

tool in the development of the VBA functions, and the error checking and interpolation improvements described in Sect. (3.3) refer to this atlas table, reason why it was opted to include it in this first version of TEOS-10 EXCEL as a supporting element for this paper. Additionally, users might have interest in ascertaining the atlas *absolute salinity anomaly* for a given location or perform further error checking, comparing TEOS-10 EXCEL output against GWS toolbox for other data sets.

### 4. Conclusions

To our knowledge, TEOS-10 EXCEL is the first implementation of the Thermodynamic Equation Of Seawater – 2010 outside the official GSW toolboxes. It does not aim, by any mean, to reproduce the full-featured GSW environment as it implements only a small subset of TEOS-10 functions. However, opening the possibility of estimating a relevant set of seawater parameters within a well-known and friendly environment (EXCEL), will hopefully democratise the compliance with current oceanographic standards among a large community of researchers and students who are not at ease with the use of high-level

programming languages. As discussed in the paper, some issues were detected with the GSW interpolation when there is missing data in the atlas reference tables. In these cases, TEOS-10 EXCEL adopts an alternative approach to the interpolation method, which has produced better results (Sect. 3.3.2). Nonetheless, this is perhaps a situation that deserves further research.





**Code and data availability**

TEOS-10 EXCEL is available for download at https://doi.org/10.5281/zenodo.4763574

**Author Contribution**

CG Martins developed the code, tested the data, and prepared the original draft. J Cross critically reviewed and edited the initial version of the manuscript.

**Competing Interests**

The authors declare that they have no conflict of interest.

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
