# Peer review of "Technical note: TEOS-10 EXCEL - Implementation of the Thermodynamic Equation Of Seawater - 2010 in EXCEL"

_Ocean Science, 2022_

## Author Comment (AC3)

**Table 1.** List of all VBA Modules and formulas included in v.2.0 of TEOS-10 Excel. Direct translations from GSW are marked with 'YES' and original or modified functions marked with 'NO'.

| VBA Module | GSW | Description |
|---|---|---|
| CT_from_pt(SA, pt) | YES | Calculates Conservative Temperature of seawater from potential temperature (whose reference sea pressure is zero dbar) |
| Entropy_part (SA, t, p) | YES | This function calculates entropy, except that it does not evaluate any terms that are functions of Absolute Salinity alone. This function is called by {pt0_from_t} |
| Entropy_part_zerop (SA, pt0) | YES | This function calculates entropy at a sea pressure of zero, except that it does not evaluate any terms that are functions of Absolute Salinity alone. This function is called by {pt0_from_t} |
| Gibbs_pt0_pt0 (SA, pt0) | YES | This function calculates the second derivative of the specific Gibbs function with respect to temperature at zero sea pressure. This function is called by {pt0_from_t} |
| Hill_ratio_at_SP2(t) | YES | Calculates the Hill ratio, which is the adjustment needed to apply for Practical Salinities smaller than 2. This function is called by {SP_from_C(C,t,p)} and {SP_from_R(R,t,p)} |
| is_Baltic(lon, lat) | NO | Checks if a location is in the Baltic Sea. This function is original and different from the GSW counterpart. Baltic limits are taken from Figure 2 of Feistel et al. (2019: 6) |
| LookUp_atlas(table_name, p, lon, lat) | NO | This function builds and interrogates the Atlas database and was developed specifically for the EXCEL implementation of TEOS-10. 'table-name' can be one of the two look-up tables [deltaSA_ref] or [SAAR_ref]. Results are a 3D interpolation of the 8 vertices of the cube around the (lon, lat, p) location in the ocean |
| pt0_from_t(SA, t, p) | YES | Calculates potential temperature with reference pressure, p_ref = 0 dbar. |
| rho(SA, CT, p) | YES | Calculates in-situ density from Absolute Salinity, Conservative Temperature, and pressure |
| SA_Baltic(SP) | YES | Calculates Absolute Salinity in the Baltic from Practical Salinity |
| sigma_CT_line(SA, sigma, min_temp, max_temp) | NO | Calculates Conservative Temperature (CT) from SA at a constant sigma value (e.g., 25) between min_temp and max_temp. Function used to build potential density (sigma) lines to be plotted in the Absolute Salinity - Conservative Temperature Diagram. It calls the {sigma0(SA,CT)} function |
| sigma0(SA, CT) | YES | Calculates potential density anomaly with reference pressure of 0 dbar |
| Sound_Speed(SA, CT, p) | YES | Calculates the speed of sound in seawater from Absolute Salinity, Conservative Temperature, and pressure |
| SP_from_C(C,t,p) | YES | Calculates Practical Salinity from Conductivity, temperature, and pressure |
| SP_from_R(R,t,p) | YES | Calculates Practical Salinity from the conductivity Ratio, temperature, and pressure |
| **Formulas used outside VBA Modules** | | |
| $t = t68 / 1.00024$ | YES | Calculates temperature ITS-90 from temperature IPTS-68 |
| $S_R = S_P * 35.16504 / 35$ | YES | Calculates Reference Salinity ($S_R$) from Practical Salinity ($S_P$) |
| $\delta S_A = S_R * $ [SAAR_Atlas] | YES | Absolute Salinity Anomaly equals the product of Reference Salinity by the interpolated Absolute Salinity Anomaly Ratio |
| $S_A = S_R + \delta S_A$ | YES | Absolute Salinity equals Reference Salinity plus Absolute Salinity Anomaly |

---

## Author Comment (AC5)

**Amended writing to account for version 2.0 upgrade.**

(Unchanged text in blue, amended text in black)

**2.1.1**

- Location: The light green tab's data template was developed to process vertical casts located at a given location. Longitude and latitude must be input in cells 'B1:B2' in decimal format (degrees). Longitude can either be within the domain (-180º to 180º) or (0º to 360º) i.e., 10º 30' W can be input as -10.5º or 349.5º. The latitude domain is (-90º to 90º) i.e., 30º S would be -30º. The input of the cast coordinates is essential, as Absolute Salinity is dependent of location (Sect 3.6). If either the longitude or the latitude cells are left empty, the Salinity Anomaly is set to zero and Absolute Salinity becomes equal to Reference Salinity.

- Pressure: pressure ($p$) units are dbar. For seawater properties, pressure is always the pressure of the water column, i.e., absolute pressure subtracted by atmospheric pressure. Therefore, at the surface, $p = 0$. For the upper ocean, 10 dbar $\approx$ 10 m.

- Salinity: the user can toggle the input between Practical Salinity (the salinity quantity which continues to be the recommended quantity to be archived (IOC, SCOR and IAPSO, 2010)), conductivity (mS cm$^{-1}$) (i.e., measured by an *in situ* transducer), or the conductivity ratio (i.e., ratio between the conductivities of the sample and of Standard Sea Water, measured by a salinometer). Column 'D' of the spreadsheet ('Practical Salinity ($S_P$)') either copies the $S_P$ value if this was the salinity input or calculates $S_P$ from conductivity using function {$SP\_from\_C(C,\ t,\ p)$} or from the conductivity ratio using function {$SP\_from\_R(R,\ t,\ p)$}, depending on the radio button selected.

- Temperature: temperature (ºC) may be selected to be either ITS-90 or IPTS-68 (data sets before 1990 are in the IPTS-68 standard, but recent data may still be applying this standard instead of the newer ITS-90 – checking the instrument specifications and/or the metadata associated with the data is advisable). Column 'J' of the spreadsheet ('Temperature ITS-90') either copies the temperature input if ITS-90 is selected or converts the IPTS-68 values to ITS-90 (ITS-90 = IPTS-68 / 1.00024). All functions use temperature ITS-90 as input.

**2.2**

- Whenever the location is in the Baltic (which is checked by the {$is\_Baltic(lon,\ lat)$} function), the Salinity Anomaly cells display 'Baltic'. This spreadsheet also includes a line with data from line one of the 'TEOS-10 Test Data' tab (surface data from the NW Pacific) as well as a line of data without location coordinates (e.g., a sample from an estuary). In this case, Salinity Anomaly is set to zero and Absolute Salinity becomes equal to Reference Salinity.

**2.6 Info tab (green)**

This tab lists all released versions of TEOS-10 EXCEL, providing detailed information on the updates included in each version.

**3. VBA (Visual Basic for Applications) modules**

**Table 1.** List of all VBA Modules and formulas included in v.2.0 of TEOS-10 Excel. Direct translations from GSW are marked with 'YES' and original or modified functions marked with 'NO'.

| VBA Module | GSW | Description |
|---|---|---|
| CT_from_pt(SA, pt) | YES | Calculates Conservative Temperature of seawater from potential temperature (whose reference sea pressure is zero dbar) |
| Entropy_part (SA, t, p) | YES | This function calculates entropy, except that it does not evaluate any terms that are functions of Absolute Salinity alone. This function is called by {pt0_from_t} |
| Entropy_part_zerop (SA, pt0) | YES | This function calculates entropy at a sea pressure of zero, except that it does not evaluate any terms that are functions of Absolute Salinity alone. This function is called by {pt0_from_t} |
| Gibbs_pt0_pt0 (SA, pt0) | YES | This function calculates the second derivative of the specific Gibbs function with respect to temperature at zero sea pressure. This function is called by {pt0_from_t} |
| Hill_ratio_at_SP2(t) | YES | Calculates the Hill ratio, which is the adjustment needed to apply for Practical Salinities smaller than 2. This function is called by {SP_from_C(C,t,p)} and {SP_from_R(R,t,p)} |
| is_Baltic(lon, lat) | NO | Checks if a location is in the Baltic Sea. This function is original and different from the GSW counterpart. Baltic limits are taken from Figure 2 of Feistel et al. (2019: 6) |
| LookUp_atlas(table_name, p, lon, lat) | NO | This function builds and interrogates the Atlas database and was developed specifically for the EXCEL implementation of TEOS-10. 'table_name' can be one of the two look-up tables [deltaSA_ref] or [SAAR_ref]. Results are a 3D interpolation of the 8 vertices of the cube around the (lon, lat, p) location in the ocean |
| pt0_from_t(SA, t, p) | YES | Calculates potential temperature with reference pressure, p_ref = 0 dbar. |
| rho(SA, CT, p) | YES | Calculates in-situ density from Absolute Salinity, Conservative Temperature, and pressure |
| SA_Baltic(SP) | YES | Calculates Absolute Salinity in the Baltic from Practical Salinity |
| sigma_CT_line(SA, sigma, min_temp, max_temp) | NO | Calculates Conservative Temperature (CT) from SA at a constant sigma value (e.g., 25) between min_temp and max_temp. Function used to build potential density (sigma) lines to be plotted in the Absolute Salinity - Conservative Temperature Diagram. It calls the {sigma0(SA,CT)} function |
| sigma0(SA, CT) | YES | Calculates potential density anomaly with reference pressure of 0 dbar |
| Sound_Speed(SA, CT, p) | YES | Calculates the speed of sound in seawater from Absolute Salinity, Conservative Temperature, and pressure |
| SP_from_C(C,t,p) | YES | Calculates Practical Salinity from Conductivity, temperature, and pressure |
| SP_from_R(R,t,p) | YES | Calculates Practical Salinity from the conductivity Ratio, temperature, and pressure |
| **Formulas used outside VBA Modules** | | |
| t = t68 / 1.00024 | YES | Calculates temperature ITS-90 from temperature IPTS-68 |
| $S_R = S_P * 35.16504 / 35$ | YES | Calculates Reference Salinity ($S_R$) from Practical Salinity ($S_P$) |
| $\delta S_A = S_R * [SAAR\_Atlas]$ | YES | Absolute Salinity Anomaly equals the product of Reference Salinity by the interpolated Absolute Salinity Anomaly Ratio |
| $S_A = S_R + \delta S_A$ | YES | Absolute Salinity equals Reference Salinity plus Absolute Salinity Anomaly |

Table 1 lists all functions (VBA modules) and formulas included in version 2.0 of TEOS-10 EXCEL. Most modules are a direct translation…

As referred before, access to the VBA project environment can be obtained by pressing [Alt + F11] (Windows) or [Fn + Alt + F11] (Mac). All functions (alphabetically listed in table 1) are described next, following the spreadsheet's column sequence.

**3.1. Practical Salinity ($S_P$)**

$S_P$ is computed from conductivity using function {$SP\_from\_C(C, t, p)$} or from the conductivity Ratio using function {$SP\_from\_R(R, t, p)$}, depending on the radio button selected. Practical Salinity is a dimensionless quantity, although PSU (Practical Salinity Units) is commonly used. For reference, the calculation algorithm is designed so that the conductivity of Reference Composition Seawater at $S_P = 35$, $t_{68} = 15$, $p = 0$ is 42.9140 mS cm$^{-1}$, which can be used to validate the conductivity function. For the conductivity Ratio function, a ratio = 1 and $t_{68} = 15$, will result in $S_P = 35$. If $S_P < 2$ both functions call the {$Hill\_ratio\_at\_SP2(t)$} function which corrects the SP value based on the Hill et al. (1986) algorithm. This algorithm is adjusted so that it is exactly equal to the PSS-78 algorithm at SP = 2.

*The following figures were updated:*

**Figure 1:** TEOS-10 EXCEL workbook light green data tab. Seawater properties in coloured columns are computed on the fly from user data pasted into white cells.

**Figure 2:** TEOS-10 EXCEL workbook 'Surface data' tab. Surface data from different locations (location coordinates for each line). Four samples are from the Baltic Sea, one from the NW Pacific and the last sample (without long/lat coordinates) is from an estuary.

[Figure]

**Figure 6:** [ndepth_ref] look-up table. The table has 45 rows (latitude) by 91 columns (longitude). South is at the top (1st row is 86° S) and 1st column is 0° of longitude. The latitude x longitude grid is a 4° x 4° grid and each cell location is obtained from the [longs_ref] and [lats_ref] tables. Cell values are the number of pressure levels at the given location. The cell highlighted in green is used as a 'case study' in the text.

| | F | G | H | I | J | K | L | M | N | O | P | Q | R | S | T | U |
|---|---|---|---|---|---|---|---|---|---|---|---|---|---|---|---|---|
| 13 | 0.007587265 | 0.008216337 | 0.007424647 | 0.005630454 | 0.0032384 | 0.001121633 | 0.000437591 | 0.000283598 | 0.000258488 | 0.000306414 | 0.000405858 | 0.000582636 | 0.001029811 | 0.001527827 | 0.001596846 | 0.001616741 |
| 14 | 0.00773597 | 0.008431254 | 0.007688479 | 0.005965611 | 0.003640463 | 0.001355465 | 0.000561454 | 0.000348497 | 0.000300451 | 0.000360793 | 0.000506601 | 0.000748773 | 0.001289316 | 0.001762179 | 0.001815071 | 0.001884277 |
| 15 | 0.007899365 | 0.008574486 | 0.007861082 | 0.006174334 | 0.003904711 | 0.001538515 | 0.000673679 | 0.000428235 | 0.000349394 | 0.000440447 | 0.000638914 | 0.000966455 | 0.001557605 | 0.001969514 | 0.002013875 | 0.0021 17049 |
| 16 | 0.008029932 | 0.008689068 | 0.008024431 | 0.006326755 | 0.004174256 | 0.001750344 | 0.000803191 | 0.000528422 | 0.000404697 | 0.000534154 | 0.000817469 | 0.001239101 | 0.001850297 | 0.002195976 | 0.002203051 | 0.002403702 |
| 17 | 0.008228679 | 0.008840657 | 0.008212917 | 0.006530541 | 0.004603277 | 0.002217398 | 0.001079402 | 0.000797616 | 0.000638354 | 0.000815376 | 0.001256342 | 0.001786972 | 0.002356018 | 0.002607539 | 0.002718048 | 0.002910265 |
| 18 | 0.008407633 | 0.008955248 | 0.008405193 | 0.006768598 | 0.004943601 | 0.002678804 | 0.001389651 | 0.001082665 | 0.000950241 | 0.001160221 | 0.001694964 | 0.002345435 | 0.002884538 | 0.00303061 | 0.003202701 | 0.003425075 |
| 19 | 0.008561275 | 0.009041577 | 0.008531257 | 0.006925568 | 0.005206232 | 0.00313519 | 0.001786091 | 0.001410546 | 0.001265549 | 0.001542952 | 0.002208032 | 0.00288733 | 0.00330817 | 0.003477958 | 0.003673256 | 0.00394365 |
| 20 | 0.008705422 | 0.009104045 | 0.008630769 | 0.00705215 | 0.00543289 | 0.003643661 | 0.00226118 | 0.001808532 | 0.001608471 | 0.002020481 | 0.002850164 | 0.003397378 | 0.003699129 | 0.003860135 | 0.004058415 | 0.004309622 |
| 21 | 0.008850281 | 0.009154356 | 0.008739148 | 0.007226648 | 0.005549866 | 0.004015798 | 0.002706156 | 0.002229445 | 0.002012965 | 0.002538207 | 0.003417199 | 0.003769656 | 0.00397721 | 0.004124866 | 0.004289072 | 0.004497948 |
| 22 | 0.008943794 | 0.009178042 | 0.008815012 | 0.007357242 | 0.005661009 | 0.004341127 | 0.003197576 | 0.002698681 | 0.002470072 | 0.002850164 | 0.003825383 | 0.003954166 | 0.004115448 | 0.00424759 | 0.004036239 | 0.004540535 |
| 23 | 0.009009886 | 0.009218692 | 0.008908996 | 0.00759247 | 0.005709105 | 0.004574747 | 0.003644211 | 0.003208049 | 0.003031175 | 0.00349246 | 0.003986676 | 0.003968637 | 0.004115448 | 0.004221737 | 0.004247794 | 0.004430649 |
| 24 | 0.009060853 | 0.009233077 | 0.008972897 | 0.007761753 | 0.005780614 | 0.004753488 | 0.004032915 | 0.003684931 | 0.003560448 | 0.003838101 | 0.003955633 | 0.003876938 | 0.003981588 | 0.004086016 | 0.004050502 | 0.004001816 |
| 25 | 0.009112921 | 0.009249478 | 0.009033516 | 0.007911187 | 0.005844102 | 0.004886662 | 0.004367845 | 0.004065591 | 0.00396472 | 0.003978199 | 0.003921243 | 0.003732477 | 0.003787497 | 0.003891735 | 0.00380428 | 0.003674715 |
| 26 | 0.009142689 | 0.0092 5642 | 0.009087358 | 0.008098981 | 0.005932269 | 0.00497313 | 0.004602549 | 0.004352101 | 0.004216718 | 0.004038171 | 0.003810867 | 0.003586259 | 0.00359505 | 0.003695226 | 0.003587182 | 0.003393917 |
| 27 | 0.009148366 | 0.009258621 | 0.009120991 | 0.008237284 | 0.006021737 | 0.005026371 | 0.004782731 | 0.00458536 | 0.004400239 | 0.00401516 | 0.003677485 | 0.003438529 | 0.003469429 | 0.00355814 3 | 0.00346893 | 0.003263141 |
| 28 | 0.009167882 | 0.009248313 | 0.009169287 | 0.008541587 | 0.006386436 | 0.005105027 | 0.004895867 | 0.004721168 | 0.004429583 | 0.003848171 | 0.003487118 | 0.003261295 | 0.003335104 | 0.003445678 | 0.003412087 | 0.003267589 |
| 29 | 0.009213968 | 0.009245437 | 0.009196107 | 0.008768696 | 0.006796146 | 0.005237766 | 0.004803792 | 0.004612022 | 0.004309108 | 0.003815922 | 0.003634408 | 0.003400565 | 0.003466424 | 0.003683605 | 0.003715926 | 0.003664202 |
| 30 | 0.009236035 | 0.009226827 | 0.009208929 | 0.008945121 | 0.007129832 | 0.005488165 | 0.004770552 | 0.004442442 | 0.004210304 | 0.003888642 | 0.003850764 | 0.003755912 | 0.0038033 | 0.004036041 | 0.004200468 | 0.00430419 |
| 31 | 0.009219045 | 0.009197872 | 0.009206341 | 0.009035997 | 0.007610069 | 0.005890486 | 0.004872129 | 0.004405444 | 0.004180367 | 0.00394193 | 0.004112428 | 0.004104743 | 0.004197571 | 0.00445 8189 | 0.004678813 | 0.004892578 |
| 32 | 0.009192553 | 0.009170229 | 0.009187315 | 0.009141968 | 0.007932154 | 0.006315794 | 0.005045773 | 0.004471019 | 0.004192354 | 0.004296012 | 0.00422 4759 | 0.004378315 | 0.004611299 | 0.004904119 | 0.005138569 | 0.005382879 |
| 33 | 0.009187974 | 0.00913718 | | | | | 0.005321122 | 0.004571465 | 0.004236636 | 0.003977235 | 0.004339081 | 0.004612237 | 0.004979521 | 0.00532738 8 | 0.005544929 | 0.005795774 |
| 34 | 0.009180469 | 0.009109531 | | | | | 0.005831923 | 0.004945996 | 0.004390707 | 0.004032286 | 0.004419669 | 0.004770676 | 0.005217486 | 0.005667592 | 0.005942029 | 0.006142602 |
| 35 | 0.009175079 | 0.009008662 | 0.009045589 | 0.00913435 | 0.008461061 | 0.00765709 | 0.0063 8616 | 0.005509364 | 0.004780258 | 0.004146816 | 0.004491119 | 0.004894749 | 0.005350943 | 0.005839766 | 0.006235628 | 0.006444394 |
| 36 | 0.009203944 | 0.009041758 | 0.008978659 | 0.009089345 | 0.008494994 | 0.007976087 | 0.007045019 | 0.006425593 | 0.005681561 | 0.004481928 | 0.004559633 | 0.004962076 | 0.005420412 | 0.005918021 | 0.006379035 | 0.006659395 |
| 37 | 0.00922741 | 0.008951321 | 0.008872803 | 0.009039996 | 0.008480566 | 0.008165775 | 0.007627993 | 0.007342053 | 0.006830766 | 0.005038608 | 0.004740746 | 0.005007445 | 0.005467756 | 0.005972721 | 0.006464604 | 0.006817884 |
| 38 | 0.009232547 | 0.008820467 | 0.008753362 | 0.009175537 | | 0.008282431 | 0.007790812 | 0.007626658 | 0.007408762 | 0.005235964 | 0.004810099 | 0.005035742 | 0.005505476 | 0.006018751 | 0.006522191 | 0.006940294 |
| 39 | 0.009110624 | 0.008768482 | 0.00867393 | | | 0.008292218 | 0.007935395 | 0.007805511 | 0.007774958 | 0.005151711 | 0.00504291 | 0.005070624 | 0.005536356 | 0.006068641 | 0.006523674 | 0.007004578 |
| 40 | 0.008806209 | 0.008739336 | 0.008706534 | | | 0.008598181 | 0.007909775 | 0.007863286 | 0.007845305 | 0.005 39689 | 0.005264025 | 0.005140008 | 0.005569826 | 0.006101186 | 0.006525859 | 0.00704963 |
| 41 | 0.008740361 | 0.008702934 | 0.008686641 | | | | 0.007908649 | 0.007888701 | 0.007852586 | | 0.005455169 | 0.005213869 | 0.005591903 | 0.006128463 | 0.006546572 | 0.00704998 |
| 42 | | 0.008659186 | 0.008640046 | | | | | | | | 0.004816461 | 0.00509936 | 0.005612582 | 0.006127909 | 0.006543907 | 0.007062829 |
| 43 | | | 0.008632253 | | | | | | | | 0.00469658 | 0.005234804 | 0.005692186 | 0.005995642 | 0.006506105 | 0.007071714 |
| 44 | | | | | | | | | | | | | | 0.005985761 | | 0.007081296 |
| 45 | | | | | | | | | | | | | | | | |
| 46 | 6 | 7 | 8 | 9 | 10 | 11 | 12 | 13 | 14 | 15 | 16 | 17 | 18 | 19 | 20 | 21 |

**Figure 8:** [deltaSA_ref] table: reference data missing for pressure levels 33 and 34 of columns 8, 9, 10 and 11.

---

## Author Comment (AC6)

Line 102:

- Salinity: the user can toggle the input between Practical Salinity (the salinity quantity which continues to be the recommended quantity to be archived (IOC, SCOR and IAPSO, 2010)), conductivity (mS cm$^{-1}$) (i.e., measured by an *in situ* transducer), or the salinometer ratio (Rt) (i.e., ratio between the conductivities of the sample and of Standard Sea Water, measured by a laboratory salinometer). Column 'D' of the spreadsheet ('Practical Salinity ($S_P$)') either copies the $S_P$ value if this was the salinity input or calculates $S_P$ from conductivity using function {*SP_from_C(C, t, p)*} or from a salinometer conductivity ratio using function {*SP_salinometer(Rt, t)*}, depending on the radio button selected. In the latter option, 't' is the temperature of the thermostable bath of the laboratory salinometer.

Line 206:

**3.1. Practical Salinity ($S_P$)**

$S_P$ is computed from conductivity using function {*SP_from_C(C, t, p)*} or from the conductivity ratio (Rt) reading of a laboratory salinometer using function {*SP_salinometer(Rt, t)*}, depending on the radio button selected. Practical Salinity is a dimensionless quantity, although PSU (Practical Salinity Units) is commonly used. For reference, the calculation algorithm is designed so that the conductivity of Reference Composition Seawater at $S_P$ = 35, $t_{68}$ = 15, $p$ = 0 is 42.9140 mS cm$^{-1}$, which can be used to validate the function. For the salinometer ratio function, a ratio = 1 will result in $S_P$ = 35, independently of the temperature. If $S_P$ < 2 both functions call the {*Hill_ratio_at_SP2(t)*} module which corrects the $S_P$ value based on the Hill et al. (1986) algorithm. This algorithm is adjusted so that it is exactly equal to the PSS-78 algorithm at $S_P$ = 2.

A VBA module to calculate Practical Salinity from the conductivity ratio (R), of a sample at temperature (t), and pressure (p) relative to SSW at t=15 °C and p=0 is also included {*SP_from_R(R, t, p)*} but it is not currently used in the template spreadsheets.

Updated Table 1

| VBA Module | GSW | Description |
|---|---|---|
| CT_from_pt(SA, pt) | YES | Calculates Conservative Temperature of seawater from potential temperature (whose reference sea pressure is zero dbar) |
| Entropy_part (SA, t, p) | YES | This function calculates entropy, except that it does not evaluate any terms that are functions of Absolute Salinity alone. This function is called by {pt0_from_t} |
| Entropy_part_zerop (SA, pt0) | YES | This function calculates entropy at a sea pressure of zero, except that it does not evaluate any terms that are functions of Absolute Salinity alone. This function is called by {pt0_from_t} |
| Gibbs_pt0_pt0 (SA, pt0) | YES | This function calculates the second derivative of the specific Gibbs function with respect to temperature at zero sea pressure. This function is called by {pt0_from_t} |
| Hill_ratio_at_SP2(t) | YES | Calculates the Hill ratio, which is the adjustment needed to apply for Practical Salinities smaller than 2. This function is called by {SP_from_C(C,t,p)} and {SP_from_R(R,t,p)} |
| is_Baltic(lon, lat) | NO | Checks if a location is in the Baltic Sea. This function is original and different from the GSW counterpart. Baltic limits are taken from Figure 2 of Feistel et al. (2019: 6) |
| LookUp_atlas(table_name, p, lon, lat) | NO | This function builds and interrogates the Atlas database and was developed specifically for the EXCEL implementation of TEOS-10. 'table-name' can be one of the two look-up tables [deltaSA_ref] or [SAAR_ref]. Results are a 3D interpolation of the 8 vertices of the cube around the (lon, lat, p) location in the ocean |
| pt0_from_t(SA, t, p) | YES | Calculates potential temperature with reference pressure, p_ref = 0 dbar. |
| rho(SA, CT, p) | YES | Calculates in-situ density from Absolute Salinity, Conservative Temperature, and pressure |
| SA_Baltic(SP) | YES | Calculates Absolute Salinity in the Baltic from Practical Salinity |
| sigma_CT_line(SA, sigma, min_temp, max_temp) | NO | Calculates Conservative Temperature (CT) from SA at a constant sigma value (e.g., 25) between min_temp and max_temp. Function used to build potential density (sigma) lines to be plotted in the Absolute Salinity - Conservative Temperature Diagram. It calls the {sigma0(SA,CT)} function |
| sigma0(SA, CT) | YES | Calculates potential density anomaly with reference pressure of 0 dbar |
| Sound_Speed(SA, CT, p) | YES | Calculates the speed of sound in seawater from Absolute Salinity, Conservative Temperature, and pressure |
| SP_from_C(C,t,p) | YES | Calculates Practical Salinity from Conductivity (mS/cm), temperature, and pressure |
| SP_from_R(R,t,p) | YES | Calculates Practical Salinity from the conductivity ratio (R), of a sample at temperature (t), and pressure (p) relative to SSW at t=15 °C and p=0 |
| SP_salinometer(Rt, t) | YES | Calculates Practical Salinity from the conductivity ratio reading of a laboratory Salinometer (Rt), where the sample and the SSW reference are at the same temperature (t). |
| **Formulas used outside VBA Modules** | | |
| $t = t68 / 1.00024$ | YES | Calculates temperature ITS-90 from temperature IPTS-68 |
| $S_R = S_P * 35.16504 / 35$ | YES | Calculates Reference Salinity ($S_R$) from Practical Salinity ($S_P$) |

| | | |
|---|---|---|
| $\delta S_A = S_R * [SAAR\_Atlas]$ | YES | Absolute Salinity Anomaly equals the product of Reference Salinity by the interpolated Absolute Salinity Anomaly Ratio |
| $S_A = S_R + \delta S_A$ | YES | Absolute Salinity equals Reference Salinity plus Absolute Salinity Anomaly |

Updated Fig.1

[Figure]

Updated Fig. 2

---

## Author Response (AR1)

**Os-2022-2**

Technical note: TEOS-10 EXCEL – Implementation of the Thermodynamic Equation Of Seawater – 2010 in EXCEL

Carlos Gil Martins and Jaimie Cross

Reply to the Reviewer's comments, Paul Barker (PB) and Richard Pawlowicz (RP), updated and corrected.

Line numbers on replies below, refer to the revised final manuscript.

**RC1, Paul Barker**

The Reviewer comments are included below in plain text, followed by the authors reply **shaded in blue**.

Review of "Technical note: TEOS-10 EXCEL – Implementation of the Thermodynamic Equation Of Seawater – 2010 in EXCEL" by Carlos Gil Martins and Jaimie Cross.

This technical note describes the implementation of TEOS-10 software in Excel. Most of the software implemented is from the GSW toolbox (McDougall and Barker, 2011), however the authors opted to adopt their own version of interpolation to compute the Absolute Salinity Anomaly Ratio that is used in the calculation of Absolute Salinity.

The note is well written and contains sufficient detail to inform the reader what it contains and how to use the software.

Many thanks for your comment, we are pleased to hear this.

General point

Note that the correct capitalisation is Absolute Salinity, Practical Salinity, Reference Salinity and Conservative Temperature.

We have now corrected this along the manuscript (capitalising and removing the italics). We applied the same to Absolute Salinity Anomaly and to Absolute Salinity Anomaly Ratio. Other physical parameters (e.g., pressure, density) were also unitalicized but not capitalised.

Detailed points.

Line 46

Delete "If"

'If' deleted and the sentence changed further, following a comment from RP. Revised sentence below (*Line 44 final version manuscript*):

As measurement technologies advance and our understanding of the oceanic environment evolves, standards relating to physical parameters frequently change in response. The definition of salinity has undergone several variations during the last century (Millero, 2010) and the temperature standard changed in 1989 from IPTS-68 to ITS-90 (Preston-Thomas, 1990). The current Thermodynamic Equation Of Seawater - 2010 (TEOS-10) has introduced a new salinity quantity, Absolute Salinity ( $S_A$ ), defined as "the mass fraction of dissolved material in seawater" (IOC, SCOR and IAPSO, 2010: 3); however, Absolute Salinity is arguably more accurately defined as the mass fraction of dissolved material in Reference Composition Seawater of the same density as that of the sample (Wright et al., 2011).

Line 54

The subscript P in  $S_P$  should not be italic.

**Corrected. All other instances of SP have also been corrected.**

Line 60 (also applies to line 11)

The estimation of Absolute Salinity in the GSW is done through a look-up table but it can be measured directly with the aid of a densimeter.

**We accept the suggestion, although we have rewritten this in a slightly different way (below), to include a further modification, driven by your following comment.**

Direct measurement of Absolute Salinity can be made with the aid of a densimeter (IOC, SCOR and IAPSO, 2010: 82), but in GSW it is estimated from interpolation of measured Absolute Salinity Anomalies stored in a world atlas look-up table. This difficulty might be a possible explanation for the absence of any previous application of TEOS-10 in EXCEL, except for a tool (GSW\_Sys\_v1.0.xlsm)1 cited in Jiang et al. (2022). That implementation of the GSW however, does not include the atlas look-up tables, using constant values of Absolute Salinity Anomaly across the major oceanic basins. We have tested this EXCEL tool, using the two data sets included in TEOS-10 EXCEL (TEOS-10 Test Data and TS-55) and, for both data sets (NW Pacific and NE Atlantic respectively), there were differences on the estimation of Absolute Salinity, starting at the 4th decimal place (positive and negative). As discussed in Sect. (3.), the results from TEOS-10 EXCEL are the same (up to 15 decimal places), for every parameter, as the ones obtained with the GSW toolbox.

Line 10 edit:

Absolute Salinity can be directly measured with the aid of a densimeter (IOC, SCOR and IAPSO, 2010: 82), but in TEOS-10 its estimation relies on the interpolation of data from casts of seawater from the world ocean (IOC, SCOR and IAPSO, 2010),

Line 61

TEOS10 for excel is included in Jiang et al (2022).

**We have addressed this on previous comment reply.**

I am not sure if the authors are aware that there is a Visual Basic version of Seawater-Ice-Air (SIA) library which includes some of the GSW functions. The SIA software is available from http://www.teos-10.org/software.htm#2

We were aware, but our initial motivation was in trying to implement the look-up tables in EXCEL, something that had not been done before. We then continued, translating the necessary functions from MATLAB. It perhaps would have been easier to use some of the above!

Lines 114– 115

The temperature acronym for temperature 68 appears as ITS-68 it should be IPTS-68.

**A typo that has been corrected (now lines 122-125).**

Line 130

If the data is not in the ocean then the Absolute Salinity value returned should be equal to the Reference Salinity, then the other values can be computed.

The updated versions of TEOS-10 EXCEL (since v.1.1) address this issue (inland/coastal locations). Now, by leaving the Longitude or Latitude cells empty, sets Absolute Salinity Anomaly to zero,  $S_A$  becomes equal to  $S_R$  and the other sample's properties are calculated.

**Line 109 update:**

If either the longitude or the latitude cells are left empty, the Salinity Anomaly is set to zero and Absolute Salinity becomes equal to Reference Salinity.

Section 2.4 SA-CT diagram

Looking at the code, I think the sigma contour that is being plotted is sigma0. Section 3.10 confirms this,  $\sigma_0$  is generally the standard notation for sigma0.

Sigma-t ( $\sigma_t = (\rho (S_A, t, 0) - 1000 \text{ kg m}^{-3}$ ) has been traditionally the standard oceanographic notation for density for a parcel of seawater not considering pressure (i.e., p = 0 dbar). The "Recommended Symbols and Units in Oceanography" (table L.1, pg. 167 of the TEOS-10 Manual) are  $\sigma_2$  for a reference pressure of 2000 dbar,  $\sigma_4$  for a reference pressure of 4000 dbar, but for p = 0 dbar it maintains the index 't', not 0, although having it as superscript ( $\sigma$ ') instead of the traditional subscript.

In the SA-CT diagram we have represented the density field with sigma calculated from Conservative Temperature instead of temperature ( $\rho$  ( $S_A$ , $\Theta$ ,0) – 1000 kg m-3) for consistency with the  $S_A$  –  $\Theta$  diagrams, and our opinion is that  $\sigma_{\Theta}$  should be the correct symbol as an indication that Conservative Temperature was used (and not temperature).

Section 3.1 SP\_from\_C

It would be great to include the low salinity (0 -2) extension to this function that is included in the GSW version of this software.

Many thanks for your suggestion. The software was updated (now version 2.2) and now includes this extension. Section 3.1 (line 224) and lines (115-121) were rewritten to accommodate this update and the upgrade of the salinity input method (radio button selection of three different 'salinity' inputs).

Section 3.3.1 Interpolation

Do you ensure that no interpolation occurs for the grids the span the Pacific Ocean and the Gulf of Mexico, across the Panama Canal?

This is not addressed by the current version but will be investigated for updating in a next version. However, leaving the Longitude or the Latitude fields empty ensures that no interpolation occurs (Salinity Anomaly is set to zero).

Section 3.3.2

It would have been great if you let the McDougall and Barker know about the missing values in the lookup table. I know that they would be eager to correct this.

Having the look-up tables as EXCEL spreadsheets turns to be a fantastic way of looking at the data and visually detect where the missing values are. This makes browsing the two look-up tables [deltaSA\_ref] and [SAAR\_ref] and locating gaps in the data much easier. The grid location of each column and depth bin (spreadsheet lines) is described in Sect. 2.5.

**References**

Jiang L-Q, Pierrot D, Wanninkhof R, Feely RA, Tilbrook B, Alin S, Barbero L, Byrne RH, Carter BR, Dickson AG, Gattuso J-P, Greeley D, Hoppema M, Humphreys MP, Karstensen J, Lange N, Lauvset SK, Lewis ER, Olsen A, Pérez FF, Sabine C, Sharp JD, Tanhua T, Trull TW, Velo A, Allegra AJ, Barker P, Burger E, Cai W-J, Chen C-TA, Cross J, Garcia H, Hernandez-Ayon JM, Hu X, Kozyr A, Langdon C, Lee K, Salisbury J, Wang ZA and Xue L (2022) Best Practice Data Standards for Discrete Chemical Oceanographic Observations. Front. Mar. Sci. 8:705638. doi: 10.3389/fmars.2021.705638

**Reference added in the required OS format:**

Jiang L.-Q., Pierrot D., Wanninkhof R., Feely R.A., Tilbrook B., Alin S., Barbero L., Byrne R.H., Carter B.R., Dickson A.G., Gattuso J.-P., Greeley D., Hoppema M., Humphreys M.P., Karstensen J., Lange N., Lauvset S.K., Lewis E.R., Olsen A., Pérez F.F., Sabine C., Sharp J.D., Tanhua T., Trull T.W., Velo A., Allegra A.J., Barker P., Burger E., Cai W.-J., Chen C.-T.A., Cross J., Garcia H., Hernandez-Ayon J.M., Hu X., Kozyr A., Langdon C., Lee K., Salisbury J., Wang Z.A., and Xue L.: Best Practice Data Standards for Discrete Chemical Oceanographic Observations. Front. Mar. Sci. 8:705638. https://doi.org/10.3389/fmars.2021.705638, 2022.

McDougall, T.J. and Barker, P.M. (2011) Getting started with TEOS-10 and the Gibbs Seawater (GSW) Oceanographic Toolbox, 28pp., SCOR/IAPSO WG127, ISBN 978-0-646-55621-5,

This reference was already included in the original manuscript.

**RC2, Paul Barker**

Reply to PB clarification was now incorporated in the answers above.

**CC1, Richard Pawlowicz**

The authors have developed an Excel spreadsheet with VBA macros that implements a small subset of the TEOS-10 software library available as the Gibbs Seawater (GSW) toolbox.

This is a great thing to have around, and something that has been on a 'wish-list' by the Joint Committee on the Properties of Seawater (JCS) for a long time.

In general, I would be happy to see this published. But I do have some concerns, as well as minor corrections.

1) My first concern is related to the fact that only a small subset of GSW is implemented.

Obviously the whole tooblox is immense and (mostly) not needed, but I suggest adding a table to the paper listing exactly which GSW functions are implemented in

macros. I could puzzle this out by opening up the macros (and sections 3.1-3.13 cover this material), but this information is then scattered over many pages.

From the outset, our objective was not to develop a full Excel implementation of the GSW toolbox. We welcome your suggestion though, and all VBA modules are now listed in Table 1.

Line 210

**Table 1.** List of all VBA Modules and formulas included in TEOS-10 Excel (v.2.1). Direct translations

 from GSW are marked with 'YES' and original or modified functions marked with 'NO'.

2) Also I note that what you apparently call 'SP\_from\_C' is not at all gsw\_SP\_from\_C. I see from your reply to an earlier comment that you want to 'leave this for the future', about which I am not really very enthusiastic. I'd rather this was fixed now, as it is unlikely that a future software update will be matched with a future documentation update like this paper.

Part of my reasoning for suggesting this get fixed now is that I can see an important audience for this EXCEL sheet are limnologists, many of whom are not very computational, but may want to see if TEOS-10 will help them. However, for them the low-frequency correction is pretty important.

The  $0 < S_P < 2$  correction has now been included (from v2.0 onwards). The former 'SP\_from\_C' function included was replaced by a direct translation of the GSW counterpart.

**Section 3.1 (line 225) was revised as follows:**

 $S_P$  is computed from conductivity using function { $SP\_from\_C(C, t, p)$ } or from the conductivity ratio (Rt) reading of a laboratory salinometer using function { $SP\_salinometer(Rt, t)$ }, depending on the radio button selected. Practical Salinity is a dimensionless quantity, although PSU (Practical Salinity Units) is commonly used. For reference, the calculation algorithm is designed so that the conductivity of Reference Composition Seawater at  $S_P = 35$ ,  $t_{68} = 15$ , p =0 is 42.9140 mS cm-1, which can be used to validate the function. For the salinometer ratio function, a ratio = 1 will result in  $S_P = 35$ , independently of the temperature. If  $S_P < 2$  both functions call the { $Hill\_ratio\_at\_SP2(t)$ } module which corrects the  $S_P$  value based on the Hill et al. (1986) algorithm. This algorithm is adjusted so that it is exactly equal to the PSS-78 algorithm at  $S_P = 2$ . A VBA module to calculate Practical Salinity from the conductivity ratio (R), of a sample at temperature (t), and pressure (p) relative to SSW (Standard Sea Water) at t=15 °C and p=0 is also included { $SP_from_R(R, t, p)$ } but it is not currently used in the template spreadsheets.

3) So, in particular, I think you need to note explicitly in the text that the practicalsalinity-from-conductivity algorithm is NOT as in GSW, and exactly how it is different.

**As per the comment above, we are confident that the practical-salinity-fromconductivity algorithm is now the same.**

4) This point is less of a concern with the work itself rather than a suggestion. I can see that one very useful target audience for this are people (e.g., lab technicians)

using salinometers, as they are much more likely to be familiar with EXCEL than with programming languages. A salinometer does not actually calculate the conductivity, instead it calculates a conductivity ratio (see gsw\_SP\_salinometer), which should then be the 'entry point' into calculations.

I urge you to consider adding this functionality - perhaps in another tab somewhat like the 'surface data' tab but which takes salinometer readings.

Thank-you for highlighting this. The Practical Salinity from the conductivity Ratio is now included.

We have updated the software to version 2.1 which now includes calculation of Practical Salinity from the conductivity ratio measured by a laboratory salinometer (Rt). The salinity input radio buttons were modified accordingly (figure below).

| Pressure (dbar) Salinity
Practical :
Conduction
Salinome | alinity
ity (mS/cm)
er Ratio (Rt)
Temperature (°C)
ITS-90
IPTS-68 |
|-------------------------------------------------------------------|----------------------------------------------------------------------------------|
|-------------------------------------------------------------------|----------------------------------------------------------------------------------|

Section 3.1 referred in a previous answer, addresses this.

5) I am not sure how difficult this is, but in many cases it is actually useful to set the salinity anomaly to zero and essentially use Reference Salinity to compute density.

In many coastal areas, for example, the look-up table is not at all accurate as it contains no information on river salts, and so it's probably better to ignore anomaly calculations completely. This is also true for inland waters. One might, for example, set up the sheets to do this if no Lat/Long is entered, or if some explicit value (e.g., 999) is entered.

This has now been fixed. Leaving the Lat/Long cells empty will set the salinity anomaly to zero, Absolute Salinity = Reference Salinity, with all other parameters estimated.

Line 109 review, referred before, reflects this update.

Minor points:

L8 "in EXCEL to estimate Absolute Salinity...."

(also note that TEOS-10 definitions like Absolute Salinity, Conservative Temperature, etc. should be capitalized - I think this is just repeating an earlier comment.)

**Yes, thank you, we have corrected this.**

L27 "to facilitate the efficient calculation of the properties...."

**Noted and corrected.**

L30-33: the GSW software is available in many programming languages other than MATLAB, so your statement is not quite correct. However, it is true that many practitioners might not be familiar with ANY programming language, so this EXCEL implement is definitely fulfilling a need.

**Thank-you for highlighting this. The text has been changed to:**

...implements the Thermodynamic Equation of Seawater – 2010 (TEOS-10) into software that calculates required seawater properties through the utilisation of programming languages (e.g., MATLAB, FORTRAN, C) that require a working understanding and knowledge of computer programming.

L42-44. It is NOT CORRECT that the properties related to the chosen variables (salinity, temperature, and pressure) must be conservative for thermodynamics to apply. In fact, they are not! It is certainly useful for numerical modelling purposes if chosen variables can be written in such a way that they are conservative under mixing, but this has nothing to do with the thermodynamic state of the fluid itself. In fact, the choice of S/T/p as state descriptions is useful in that these are measurable variables and are natural for a Gibbs function description of the state.

This sentence should probably be removed.

**Thank-you for correcting this. The sentence in question has been removed from the manuscript.**

L46: this is a little trivial, but temperature standards have also changed over time - before IPTS-68 there were a number of other standards - so it is not true that the concept has remained 'unaltered over time'.

**Line 44:**

As measurement technologies advance and our understanding of the oceanic environment evolves, standards relating to physical parameters frequently change in response. The definition of salinity has undergone several variations during the last century (Millero, 2010) and the temperature standard changed in 1989 from IPTS-68 to ITS-90 (Preston-Thomas, 1990).

L47-48. Technically, absolute salinities (lower case) are mass fraction definitions. However, Absolute Salinity S\_A (capitalized as defined in TEOS-10) is actually "the mass fraction of dissolved material in Reference Composition Seawater of the same density as that of the sample" (it is in fact a density salinity). Yes, this is complicated and confusing.

**Thank-you for pointing this out. We have incorporated this correction into the revised sentence, as follows:**

The current Thermodynamic Equation Of Seawater - 2010 (TEOS-10) has introduced a new salinity quantity, Absolute Salinity ( $S_A$ ), defined as "the mass fraction of dissolved material in seawater" (IOC, SCOR and IAPSO, 2010: 3); however, Absolute Salinity is arguably more accurately defined as the mass fraction of dissolved material in Reference Composition Seawater of the same density as that of the sample (Wright et al., 2011).

Wright, D. G., Pawlowicz, R., McDougall, T. J., Feistel, R., and Marion, G. M.: Absolute Salinity, "Density Salinity" and the Reference-Composition Salinity Scale: present and future use in the seawater standard TEOS-10, Ocean Sci., 7, 1–26, https://doi.org/10.5194/os-7-1-2011, 2011.

L53: S\_A and CT are natural arguments into a simpler and more computationally efficient 75-term density/specific volume equation by Roquet et al. (2015), which can be used to derive some (but not all) other thermodynamic properties, although the best reference is still the TEOS-10 Gibbs function.

Our intention is to inform that SA and CT are the arguments to be used in the estimation of the other thermodynamic properties. As such, we are confident that the following sentence adequately explains this:

These two new quantities,  $S_A$  and  $\Theta$ , together with pressure (p), are now the arguments of the equation of state, and to compute any thermodynamic property of seawater (e.g., density, sound speed) they must be estimated first.

L101: Maybe clarify that pressure is "sea pressure, i.e. absolute pressure - 10.1325 dbar".

**Line 111 update:**

Pressure: pressure (p) units are dbar. For seawater properties, pressure is always the pressure of the water column, i.e., absolute pressure subtracted by atmospheric pressure. Therefore, at the surface, p = 0. For the upper ocean, 10 dbar ≈ 10 m.

L203: the [Alt-F11] is probably a Windows-specific command. It doesn't work for Excel on Macs.

**This is also referred in Line 77.**

**Line 84:**

Pressing [Alt + F11] (Windows) or [Fn + Alt + F11] (Mac) opens the VBA environment allowing access to the 15 function modules (table 1), although access to these is not required to make use of the Workbook, nor is a working knowledge of VBA.

**Line 220:**

As referred before, access to the VBA project environment can be obtained by pressing [Alt + F11] (Windows) or [Fn + Alt + F11] (Mac). All functions (alphabetically listed in table 1) are described next, following the spreadsheet's column sequence.

L210: It is not exactly true that the conductivity of SSW at standard conditions is 42.9140 mS/cm. This is in fact a number which a) has never been replicated, and b) is more accurate than we can realistically measure conductivity. So, it is perhaps more true to say that the software is designed so that by definition the conductivity of Reference Composition Seawater with S\_P=35 is 42.9140 mS/cm at standard conditions.

**Thank-you for this observation. We agree, so the sentence has been rephrased to (Line 226)**

For reference, the calculation algorithm is designed so that the conductivity of Reference Composition Seawater at  $S_P = 35$ ,  $t_{68} = 15$ , p = 0 is 42.9140 mS cm-1, which can be used to validate the function.

**Further updates to the final manuscript, not mentioned in the answers above**

**Line 122:**

• Temperature: temperature (°C) may be selected to be either ITS-90 or IPTS-68 (data sets before 1990 are in the IPTS-68 standard, but recent data may still be applying this standard instead of the newer ITS-90 – checking the instrument specifications and/or the metadata associated with the data is advisable). Column 'J' of the spreadsheet ('Temperature ITS-90') either copies the temperature input if ITS-90 is selected or converts the IPTS-68 values to ITS-90 (ITS-90 = IPTS-68 / 1.00024). All functions use temperature ITS-90 as input.

**New section 2.6**

**2.6 Info tab (green)**

This tab lists all released versions of TEOS-10 EXCEL, providing detailed information on the updates included in each version.

**Figures 1, 2, 6 and 8 were updated**

**The DOI for the software was updated (line 13, 371)**

Line 371

TEOS-10 EXCEL is available for download at https://doi.org/10.5281/zenodo.4748829. This DOI represents all versions and will always resolve to the latest one. This manuscript reflects the status of version 2.1.

Line 374

CG Martins developed the code, tested the data, and prepared the original draft. J Cross critically reviewed and edited the initial and final versions of the manuscript.

Several other writing corrections (grammar) were made along the final manuscript

- END -